# InfoDisent: Explainability of Image Classification Models by Information Disentanglement

## Abstract

In this work, we introduce InfoDisent, a hybrid approach to explainability based on the information bottleneck principle. InfoDisent enables the disentanglement of information in the final layer of any pretrained model into atomic concepts, which can be interpreted as prototypical parts. This approach merges the flexibility of post-hoc methods with the concept-level modeling capabilities of self-explainable neural networks, such as ProtoPNets. We demonstrate the effectiveness of InfoDisent through computational experiments and user studies across various datasets using modern backbones such as ViTs and convolutional networks. While InfoDisent achieves competitive performance within the class of interpretable models, we observe an accuracy-interpretability trade-off when compared to black-box counterparts, especially visible in CNNs. Notably, InfoDisent generalizes the prototypical parts approach to novel domains (ImageNet).

## 1 Introduction

Deep neural networks have demonstrated performance that matches or even surpasses human capabilities across various domains, such as image classification and generation, speech recognition, and natural language processing. Despite their impressive achievements, these networks often operate as "black boxes", offering little insight into the reasoning behind their decisions Rudin (2019). This lack of transparency poses significant challenges, particularly in high-stakes applications such as medicine and autonomous driving, where understanding a model's decision-making process is critical Bojarski et al. (2017); Khan et al. (2001); Nauta et al. (2023b); Patrício et al. (2023); Samek et al. (2021); Struski et al. (2024). To address this issue, the subfield of artificial intelligence known as eXplainable AI (XAI) Xu et al. (2019) has emerged, focusing on making AI systems more interpretable.

XAI methods can be classified into two major categories: post-hoc methods and inherently interpretable (ante-hoc) methods Rudin (2019). Post-hoc methods, such as GradCAM Selvaraju et al. (2017), are highly flexible because they can be applied to any neural network architecture. However, they are often unreliable Adebayo et al. (2018b), provide local explana-

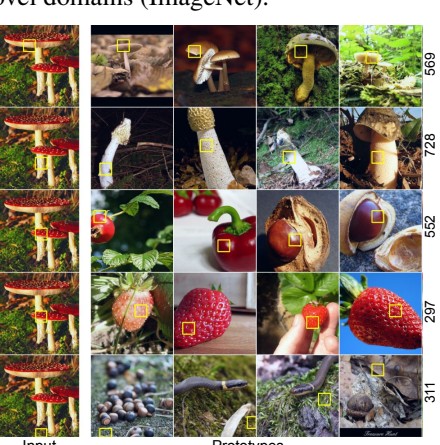

Figure 1: Decision explanation constructed by InfoDisent for the pre-trained ViT feature space on the *Agaric* mushrooms image from the ImageNet. We can trace the decision of ViT behind assigning the class *Agaric* to the image on the left to having a hat (569), a white leg (728), a reddish shine (552), a strawberry texture (297) and the appearance of ground with moss (311). Note that,Each row in the prototype block represents a specific prototypical part (channel number), and the first column shows their activation in the original image.

tions, and fail to reveal the characteristics of specific classes. In contrast, ante-hoc methods, such as ProtoPNet Chen et al. (2019), offer concept-level explanations by identifying prototypical parts from the training dataset. While this approach enhances interpretability, it is limited to fine-grained classification tasks and requires backbone fine-tuning in the case of ProtoPNet. Consequently, these

methods require substantially computational power to train, especially when backbones such as Vision Transformers (ViTs) Dosovitskiy et al. (2020) are fine-tuned in ProtoViT Ma et al. (2024).

To address this, we propose InfoDisent, a novel XAI approach that leverages information disentanglement in the final layer of any pretrained model. In InfoDisent, each channel encapsulates a single atomic concept, interpretable as prototypical parts – similar to the approach used in PIPNet Nauta et al. (2023a). This is achieved through the application of an information bottleneck, in which we enforce activation sparsity in each prototypical channel. We decided to use orthoganilization to enforce sparsity, as this mechanism can be successfully applied to ViTs Huang et al. (2022); Tang et al. (2022) and CNNs Wang et al. (2020). As a result, InfoDisent provides both local and global-level explanations in the form of human-interpretable concepts, as illustrated in Fig. 1. Furthermore, InfoDisent offers significant flexibility, as it can be seamlessly applied to any pretrained architecture, including ViTs and convolutional networks.

We demonstrate the effectiveness of InfoDisent through both computational experiments and user studies. Our method is benchmarked on five datasets, including ImageNet, a challenging dataset where prototypical parts-based approaches have not been generalized to. We observe that InfoDisent achieves competitive performance within the class of interpretable models, we observe an accuracy-interpretability trade-off when compared to black-box counterparts, especially visible in CNNs. Additionally, results from the user study indicate that InfoDisent provides a competitive level of explanation understanding compared to other prototypical parts-based methods, while offering the advantage of flexibility in its application to any model backbone. We make the code available.

Our contributions can be summarized as follows:

- We propose InfoDisent, a hybrid XAI model that combines the interpretability of ante-hoc methods with the flexibility of post-hoc approaches.
- InfoDisent's unique advantage is providing prototypical part-like interpretations for the feature space of any pretrained network without modification or retraining of a backbone.
- We validate the effectiveness of InfoDisent in terms of both accuracy and user understanding through extensive experimental evaluations.
- With InfoDisent, we generalize prototypical parts beyond fine-grained classification, which is a major limitation of existing methods Elhadri et al. (2025).

## 2 RELATED WORKS

Research in XAI can be divided into two disjoint categories: post-hoc interpretability Lundberg & Lee (2017); Ribeiro et al. (2016); Selvaraju et al. (2017), where we analyze the pre-trained model to explain its predictions, and inherently explained models Böhle et al. (2022); Chen et al. (2019), where the aim lies in building networks which decisions are easy to interpret. Both of the above approaches have their advantages and disadvantages which we discuss in the following paragraphs.

**Post-hoc methods** In the post-hoc methods, we interpret existing pre-trained network architectures. The commonly used methods such as SHAP Lundberg & Lee (2017); Shapley (1951), LIME Ribeiro et al. (2016), LRP Bach et al. (2015) and Grad-CAM Selvaraju et al. (2017) provide in practice only feature importance whic can be visualized as a saliency map that shows on which part of the image the model has focused its attention. This allows us to check if the model does not focus its attention outside of the object of interest Ribeiro et al. (2016), however, it is in general not sufficient to really understand the reasons behind given predictions. Additionally, post-hoc methods allow typically only local explanations (per the prediction of a given image), and do not allow to understand of the prerequisites to the given class.

**Inherently explained models** While post-hoc methods are easy to implement due to their non-intrusive nature, they often produce biased and unreliable explanations Adebayo et al. (2018a). To address this, recent research has increasingly focused on designing self-explainable models that make the decision process directly visible Brendel & Bethge (2019); Alvarez Melis & Jaakkola (2018). Many of these interpretable solutions utilize attention mechanisms Liu et al. (2021); Zheng et al. (2019) or exploit the activation space, such as with adversarial autoencoders Guidotti et al. (2020). Among the most recent approaches, two of the approaches has significantly influent the development of self-explainable models, which are Concept Bottlenecks Koh et al. (2020) and Pro-

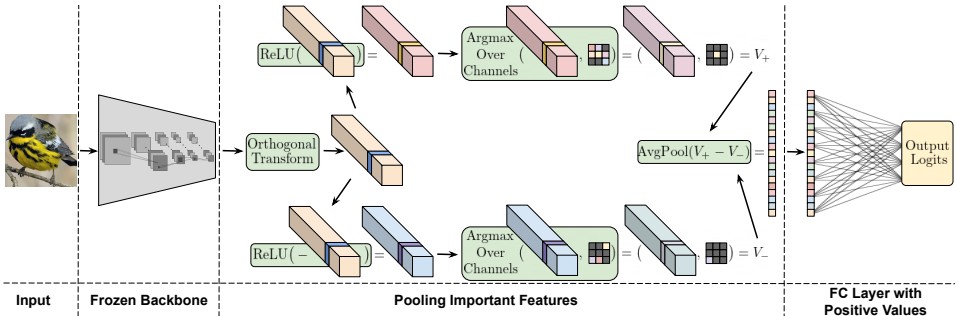

Figure 2: The architecture used for training of our proposed image classification interpretation model. InfoDisent is composed of three main components: a pre-trained backbone, a pooling layer for extracting important features, and a fully connected layer. The backbone is a pre-trained CNN or transformer with frozen weights, meaning it is not further trained. In the initial pooling layer, the model extracts representations from the last convolutional layer of the backbone and identifies key features within each channel, targeting both positive and negative activations through the application of the $\arg\max$ operation. However, during training, we replace the $\arg\max$ operation with the Gumbel-Softmax trick, which achieves a similar outcome in a differentiable manner. In the next step, these positive and negative features are pooled at the channel level to create a dense vector, where the vector's dimensions correspond to the number of channels. Finally, this dense vector is passed through a fully connected linear layer with positive weights in the network's final component. toPNet Chen et al. (2019). Concept Bottleneck originaly learns to predict in a supervised fashion which concept are present on the image, and then decision is made based on that representation. Among others, this idea has been further developed by detecting the concept in an unsupervised way Hu et al. (2025); Rao et al. (2024). ProtoPNet learns class-specific prototypes, similar to concepts, with a fixed number per class. The model classifies inputs by calculating responses from each class's prototypes and summarizing these responses through a fully connected layer, providing explanations as a weighted sum of all prototypes. This method inspired the development of several other self-explainable models Donnelly et al. (2022); Nauta et al. (2023a); Pach et al. (2024); Rymarczyk et al. (2021; 2022; 2023); Wang et al. (2021). Typically, in the prototypical parts-based model, the decision of a given class is decomposed into the appearance of a few selected prototypes, which are similar to some strongly localized parts of some chosen images from the training data.

## 3 INFODISENT

Our approach (presented in Fig. 2) is inspired by the core principles of prototypical models, which aim to ground classification decisions in the co-occurrence of localized, visualizable prototypes within an image. Specifically, these models typically: (i) attribute the final class prediction to the presence of specific prototypes in the input; (ii) allow for the interpretation of these prototypes through their correspondence to training examples; and (iii) ensure that prototypes represent spatially constrained, meaningful image parts.

We hypothesize that even within the feature representations of pre-trained models, it is possible to identify and isolate channels that inherently possess those properties. To achieve this, InfoDisent introduces a mechanism for disentangling the channels in the feature space. Specifically, we apply an orthogonal transformation directly in the pixel space of the feature maps. This operation preserves the inner-lying distance and scalar products, ensuring that the expressiveness of the representation of the pre-trained model is maintained.

By focusing on disentangling the feature channels, our method operates directly on the feature maps produced by the backbone network. Consequently, we consider a dataset of feature space images, characterized by potentially varying spatial resolutions but a consistent number of channels $d$.

**Default classification head in deep networks.** To establish notation, let us first describe the typical classification head. In the case of a classification task with $k$ classes, we apply the following operations for the image $I$ in the feature space:

1. $I \to v_I = \text{avg\_pool\_over\_channels}(I) \in \mathbb{R}^d$

2. $v_I \rightarrow w_I = Av_I$, where $A$ is a matrix of dimensions $d \times k$

3. $w_I \rightarrow p_I = \mathrm{softmax}(w_I)$.

In the case of InfoDisent we apply a few mechanisms to ensure the desired properties. First, we need to be able to disentangle the channel space. To do so we apply the unitary map $U$ in the pixel space. Next, we apply the information bottleneck – for a given channel instead of the average pool where all pixels participate, we use extremely sparse analog where only the value of the highest positive and negative pixels are used. Finally, as is common in interpretable methods we use matrix $A$, but only with nonnegative coefficients to allow only positive reasoning.

**The sparse pooling features mechanism**  Most interpretable models involve retraining certain parts of the CNN Chen et al. (2019); Rymarczyk et al. (2022), whereas others, like PIP-Net Nauta et al. (2023a), retrain the entire CNN. In contrast, our method utilizes a pre-trained CNN or transformer without further modification during training. We first employ a trainable orthogonal transformation $U$ on pixel space to enable the disentanglement of hidden features from feature maps[1]. To enforce the disentanglement we follow by an introduced sparse analogue of average pooling over channel $K$ given by

$$K \rightarrow \mathrm{mx\_pool}(K) = \max(\mathrm{ReLU}(K))$$
$$- \max(\mathrm{ReLU}(-K)).$$

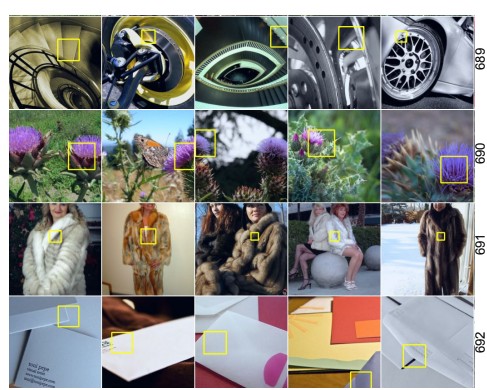

Figure 3: The image shows prototypes from channels 689 to 692 in a trained ResNet-50 on the ImageNet. Each row displays the 5 most significant patches from a single prototypical channel. The prototype's activations are highlighted by yellow boxes.

Observe that to compute $\mathrm{mx\_pool}(K)$ we need to know only the highest positive and negative pixel values of $K$, contrary to $\mathrm{avg\_pool}$, where all the pixel values are needed. Subsequently, we identify sparse representations (superpixels) within the channels that contribute positively or negatively to predictions. This enables us to generate heatmaps similar to Grad-CAM Selvaraju et al. (2017) without necessitating a backward model step, as shown in Fig. 4a. Importantly, unlike Grad-CAM, our technique supports the visualization of negative heatmaps, resembling the LRP Bach et al. (2015) method that requires a backward pass in a neural network. Our method operates solely during the forward step. Finally, as is common in XAI models, to allow only positive reasoning we allow the matrix $A$ to have only nonnegative values.

**Classification head in InfoDisent.**  Finally, the classification head in InfoDisent is given by:

1. $I = (I_{rs})_{rs} \rightarrow J = (UI_{rs})$, where $U : \mathbb{R}^d \rightarrow \mathbb{R}^d$ is an orthogonal matrix and $I_{rs}$ denotes the pixel value of $I$ with coordinates $r$ and $s$,

2. $J \rightarrow v_J = \mathrm{mx\_pool\_over\_channels}(J) \in \mathbb{R}^d$, where for a given channel $K$ we have

$$\mathrm{mx\_pool}(K) = \max(\mathrm{ReLU}(K)) - \max(\mathrm{ReLU}(-K)),$$

3. $v_J \rightarrow w_J = Av_J$, where $A$ is a matrix with nonnegative coefficients of dimensions $d \times k$

4. $w_J \rightarrow p_I = \mathrm{softmax}(w_J)$

**InfoDisent model.**  Thus InfoDisent consists of two main components: the frozen CNN or transformer Backbone, and InfoDisent classification head, as illustrated in Fig. 2.

The first component is a backbone, CNN or ViT, which is a frozen pre-trained network up to the final layer that generates the last feature map. Importantly, this part of the network and its weights

---

[1]To parametrize orthogonal maps, we restrict to those with positive determinants and use the formula $U = \exp(W - W^T)$, where $\exp(\cdot)$ denote the matrix exponential Hall & Hall (2013). We utilize the fact that the space of orthogonal matrices with positive determinants coincide with exponentials of skew-symmetric matrices Shepard et al. (2015).

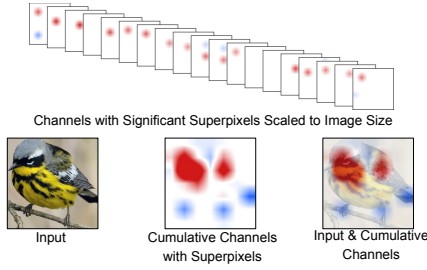

Channels with Significant Superpixels Scaled to Image Size

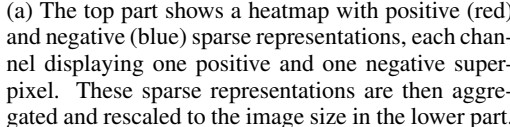

Input     Cumulative Channels with Superpixels     Input & Cumulative Channels

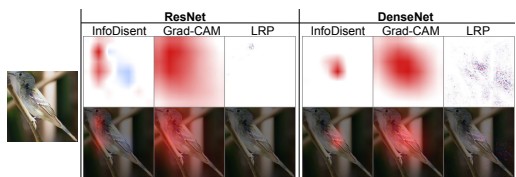

(a) The top part shows a heatmap with positive (red) and negative (blue) sparse representations, each channel displaying one positive and one negative superpixel. These sparse representations are then aggregated and rescaled to the image size in the lower part.

(b) Example compares visual explanations from InfoDisent, Grad-CAM, and LRP. On the left we show the input, with the columns displaying consecutive explanation types for ResNet and DenseNet.

Figure 4: The image demonstrates how to analyze and visualize decisions made by InfoDisent.

remain unaltered during the training of InfoDisent, ensuring that the learned feature representations are preserved. The second component involves pooling important features from the final feature maps produced by the frozen backbone. This pooling mechanism encourages the network to use information bottlenecks to build maximally informative channels independent of each other. This leads to constructing prototypical channels, which can be easily interpreted. The final element is the fully connected layer, which consists of only positive weight values. This restriction on positivity, except biases handled as in conventional linear layers, ensures that the information about the positive or negative contributions of selected features from the previous part of the model is preserved. This constraint is beneficial for the interpretability of the model's predictions, as it clarifies the contribution of each feature to the final output. Observe, that contrary to some of the post-hoc methods, InfoDisent needs training of the model on the whole dataset.

**Gumbel-Softmax** To maximize the extraction of information from prototypical channels, we introduce an information bottleneck within our model architecture. This is achieved by applying the $\arg\max$ operation to individual channels, see Fig. 2. While the $\arg\max$ function can be used to extract a sparse representation from feature maps, our goal is to enable the model to learn to select the most important values. To achieve this, we require a differentiable $\arg\max$ function. The ideal solution for this is the Gumbel-Softmax estimator Jang et al. (2016). Given $x = (x_1, \ldots, x_D) \in \mathbb{R}^D$ and $\tau \in (0, \infty)$,

$$\text{Gumbel-Softmax}(x, \tau) = (y_1, \ldots, y_D) \in \mathbb{R}^D,$$

where

$$y_i = \frac{\exp\left((x_i + \eta_i)/\tau\right)}{\sum_{d=1}^{D} \exp\left((x_d + \eta_d)/\tau\right)},$$

and $\eta_d$ for $d \in \{1, \ldots, D\}$ are samples taken from the standard Gumbel distribution.

The Gumbel-Softmax distribution serves as an interpolation between continuous categorical densities and discrete one-hot encoded categorical distributions, with the discrete form being approached as the temperature $\tau$ decreases within the range of $[0.1, 0.5]$. In our experiments, we initialized $\tau$ at 1 and progressively reduced it to 0.2. Finally, at the end of the training, we applied a hard softmax.

Following the extraction of key features using the sparse operation – specifically, the $\arg\max$ operation via the Gumbel-Softmax trick – we preserve the original structure of the network's output, maintaining the classical form of the convolutional network's output, as shown in Fig. 2. During the subsequent aggregation of positive and negative features, we utilize an average pooling operation to consolidate the information. This approach ensures that the pooled features capture a balanced representation of the activations, contributing to a robust final output.

## 3.1 UNDERSTANDING THE CLASSIFICATION DECISIONS

**Prototypes in InfoDisent.** The main feature of InfoDisent is its ability to disentangle the channels making them interpretable. Thus, similarly to PiP-Net Nauta et al. (2023a), we identify channels as prototypes. To illustrate a given prototype channel, we present five images from the training dataset on which the activation of the channel is the greatest, see Fig. 3, where we present consecutive prototypes for the pre-trained ResNet-50 model. This follows the fact that for better interpretability of model decisions, it is beneficial for humans to be presented from 4 to 9 concepts Rymarczyk et al. (2022). Formally, similarly as in prototypical models, as the prototypical part, we understand the part of the image corresponding to pixels in feature space with maximal activation, marked by the yellow box in Fig. 3. Observe that the presented prototypes seem consistent with each other, and could be well interpreted.

**Understanding the Decision for a Given Image by Prototypes.** Now that we can understand and visualize prototypes, there appears to be a question of how to visualize crucial prototypes from the model's decision perspective for a given image. To do this we chose 5 prototypical channels which are most important for the prediction[2] in Fig. 5. For each channel, we identified 5 images from the training dataset that exhibit the strongest activation values for that channel, as depicted by the red spots in Fig. 4a. Simply, we selected the top 5 images based on the highest activation values, or $\arg\text{-top}5$, for each channel. The model's proposed prototypes are easy to interpret. Moreover, unlike current state-of-the-art prototype methods, our model excels at interpreting images from the ImageNet dataset. More examples from various datasets and models are presented in the Supplementary Materials.

**Heatmaps.** Our approach, which relies on representation channels, enables us to easily generate heatmaps similar to those produced by the Grad-CAM method, see Fig. 4a. To do this we accumulate the activations of all prototypes (both positive and negative ones) over all channels. Since in InfoDisent we use information bottleneck, we obtain more localized results than other standard approaches.

Observe that, unlike Grad-CAM, our heatmaps also illustrate negative activations, similarly to the Layer-Wise Relevance Propagation (LRP). While LRP effectively highlights both positive and negative contributions, it can be complex to implement and sensitive to changes in model architecture. Our approach, by using a non-negative last layer and a prototypical-parts-based architecture, directly addresses these issues. This design results in clear and interpretable visualizations (see Fig. 23) that can be compared with those produced by methods such as GradCAM and LRP.

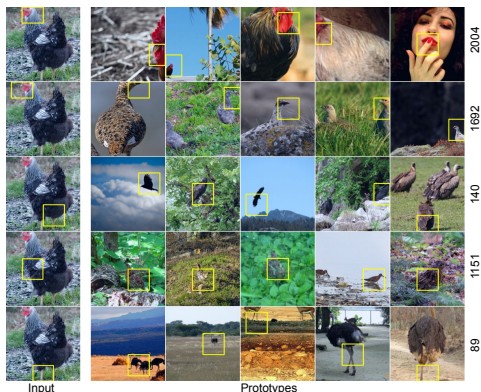

Figure 5: Exemplary explanation (*hen*) for ResNet-50 backbone provided by InfoDisent in a form of prototypical parts.

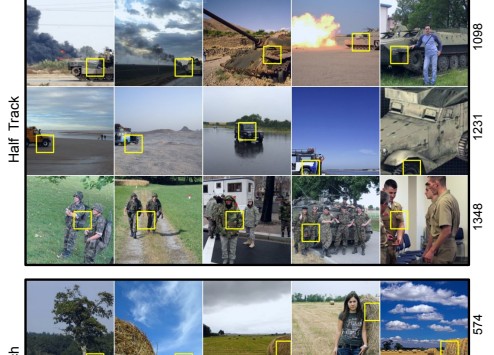
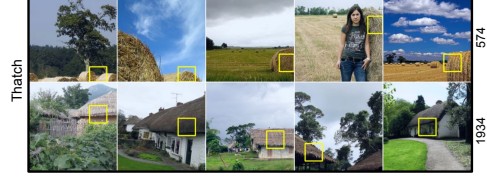

Figure 6: Prototypes of two classes (from top to bottom): Half Track and Thatch from the ImageNet dataset. Each row displays the 5 most significant prototypes for a specific channel, with the channel numbers listed on the right. The order of the prototypes within each row reflects their importance for explaining the given class. In the Half Track class, the prototypes from channel 1098 are the most crucial, followed by those from channels 1231 and 1348. These prototypes effectively explain the Half Track class, as they highlight elements of a tank (from channel 1098), a car, and the possible presence of soldiers on board. Note that the prototypes are all generated at the same resolution. If they appear to be of different sizes across images, it's because the original test images themselves varied in resolution.

---

[2]One can assume more subtle strategies, see Supplementary Materials.

**Understanding the Decision Behind Class.** We examine the model's decision-making process on a per-class basis by utilizing prototypes. To identify prototypes for a given class, we focus on key channels that are prominently activated across all test set images belonging to that class. These key channels are selected based on their consistent presence and strong activation in images of the same class. Once we have identified these crucial channels, we visualize the prototypes as previously described, providing a clear representation of what the model finds important for a given class. Fig. 6 illustrates the prototypes for selected classes from the ImageNet dataset. Each prototype captures essential features such as material types, structural elements, or specific textures that are related to the class.

## 4 NUMERICAL EXPERIMENTS

This section outlines the experiments conducted to compare our approach with current state-of-the-art methods, highlighting the most significant results. Our experiments utilized a variety of datasets, including well-established benchmarks for evaluating the explainability of artificial models: The Caltech-UCSD Birds-200-2011 (CUB-200-2011) Wah et al. (2011), Stanford Cars Krause et al. (2013), and Stanford Dogs Khosla et al. (2011). Additionally, we incorporated the full ImageNet Russakovsky et al. (2015) dataset for the first time in the class of prototypical networks. Additional results, including ablation study and analysis of inter-channel correlation, and details concerning those experiments are provided in the Supplementary Materials. Those experiments show that InfoDisent consistently achieves superior or mathing Disentanglement metrics in most of the cases, demonstrating high concept coherence with minimal variability between data-classes of a concept. Furthermore, we show that InfoDisent yields robust prototypical parts and outperforms ProtoViT on 2 out of 3 metrics related to spatial misalignment. We conclude by presenting qualitative examples and additional detailed results that complement those from the main body.

### 4.1 CLASSIFICATION PERFORMANCE

To compare our approach, we selected several state-of-the-art, interpretable models based on the same CNN architectures as our approach. We categorized the models into a few groups based on their CNN architecture – ResNet-34/50 He et al. (2016), DenseNet-121 Huang et al. (2017), and ConvNeXt-Tiny Liu et al. (2022b) – and the experiments we conducted on these datasets. Specifically, we performed two experiments: the first used cropped images for training and testing, and the second used full images.

In the first experiment, we utilized two key datasets: CUB-200-2011 and Stanford Cars, which are frequently employed in prototype model evaluations. We trained both the base models and InfoDisent on these cropped images, with the results shown in Tab. 1. In the second experiment, we used CUB-200-2011 and Stanford Dogs datasets, but this time with the full images. The results of this experiment are detailed in Tab. 2.

In both experiments, our approach consistently outperformed or matches the compared black-box models. When compared to interpretable models, InfoDisent's performance varied by backbone, achieving comparable or slightly lower results than state-of-the-art models, especially with ViT. This minor difference is attributable to the competing interpretable models, which modify and fine-tune the backbone during training. This adaptation allows them to specialize a generally-trained backbone for the fine-grained task, which InfoDisent, being non-modifying the backbone, does not benefit from. Crucially, however, these backbone modifications require significantly greater computational resources and more complex training procedures. In sharp contrast, our method simplifies the training process by only adjusting the last two parts of the model while leaving the original backbone unchanged, leading to a substantial reduction in both required resources and training time (see Tab. 8).

In the next experiment, we utilized the full ImageNet dataset and evaluated both traditional CNN models – such as ResNet-34/50, DenseNet-121, and ConvNeXt-Large – and popular transformer models, including VisionTransformer (ViT-B/16) Dosovitskiy et al. (2020), SwinTransformer (Swin-S) Liu et al. (2022a), and MaxVit Tu et al. (2022). Given that current prototype models did not perform well on the ImageNet dataset and thus lack evaluation results, Tab. 3 presents a comparison between the classical models and our approach. Typically, prototype models exhibit

lower performance on more complex datasets compared to traditional models as shown in Tab. 3, but it also offers enhanced explainability. This difference with accuracy we identify as accuracy-interpretability trade-off. We hypothesize that the orthogonal disentanglement enforces a "part-based" sparsity that conflicts with the distributed, texture-biased representations of ResNets Hermann et al. (2020), whereas ViTs (which process patches globally) have better adaptability capabilities Naseer et al. (2021).

Table 1: Accuracy comparison of interpretability models using standard CNN architectures (utilized in explainable models) trained on cropped bird images of CUB-200-2011, and Stanford Cars (Cars). Our approach demonstrates superior performance across nearly all the datasets and models considered. For each dataset and backbone, we boldface the best result in the class of interpretable models.

| Model | Dataset | |
| --- | --- | --- |
| | CUB-200-2011 [%] | Cars [%] |
| ResNet-34 | 82.4 | 92.6 |
| ↳InfoDisent (ours) | **83.5 ± 0.02** | **92.8 ± 0.04** |
| ProtoPNet | 79.2 ± 0.3 | 86.1 ± 0.2 |
| ProtoPShare | 74.7 | 86.4 |
| ProtoPool | 80.3 ± 0.2 | 89.3 ± 0.1 |
| ST-ProtoPNet | **83.5 ± 0.2** | 91.4 ± 0.3 |
| TesNet | 82.7 ± 0.2 | 90.9 ± 0.3 |
| ResNet-50 | 83.2 | 93.1 |
| ↳InfoDisent (ours) | **83.0 ± 0.07** | **92.9 ± 0.02** |
| ProtoPool | – | 88.9 ± 0.1 |
| ProtoTree | – | 86.6 ± 0.7 |
| PIP-Net | 82.0 ± 0.2 | 86.5 ± 0.3 |
| DenseNet-121 | 81.8 | 92.1 |
| ↳InfoDisent (ours) | 82.6 ± 0.02 | **92.7 ± 0.02** |
| ProtoPNet | 79.2 ± 0.3 | 86.8 ± 0.1 |
| ProtoPShare | 74.7 | 84.8 |
| ProtoPool | 73.6 ± 0.2 | 86.4 ± 0.3 |
| ST-ProtoPNet | **85.4 ± 0.1** | 92.3 ± 0.2 |
| TesNet | 84.8 ± 0.2 | 92.0 ± 0.3 |
| ConvNeXt-Tiny | 83.8 | 91.0 |
| ↳InfoDisent (ours) | 84.1 ± 0.08 | **90.2 ± 0.01** |
| PIP-Net | **84.3 ± 0.2** | 88.2 ± 0.5 |
| DeiT-Small | 84.3 | – |
| ↳InfoDisent (ours) | 83.7 ± 0.03 | – |
| ProtoViT | **85.4 ± 0.5** | 91.8 |

Table 2: Classification accuracy on full CUB-200-2011, and Stanford Dogs datasets by competing approaches using different CNN backbones. For each dataset and backbone, we boldface the best result in the class of interpretable models.

| Model | Dataset | |
| --- | --- | --- |
| | CUB-200-2011 [%] | Dogs [%] |
| ResNet-34 | 76.0 | 84.5 |
| ↳InfoDisent (ours) | **78.3 ± 0.15** | **83.9 ± 0.21** |
| ProtoPNet | 74.1 ± 0.2 | 76.1 ± 0.3 |
| ST-ProtoPNet | 78.2 ± 0.1 | 83.4 ± 0.5 |
| TesNet | 76.5 ± 0.1 | 81.2 ± 0.3 |
| ResNet-50 | 78.7 | 87.4 |
| ↳InfoDisent (ours) | 79.5 ± 0.56 | **86.6 ± 0.25** |
| ProtoPNet | 84.8 ± 0.4 | 78.1 ± 0.3 |
| ST-ProtoPNet | **88.0 ± 0.2** | 83.3 ± 0.3 |
| TesNet | 87.3 ± 0.2 | 85.7 ± 0.4 |
| DenseNet-121 | 78.2 | 84.1 |
| ↳InfoDisent (ours) | 80.6 ± 0.37 | **83.8 ± 0.07** |
| ProtoPNet | 76.6 ± 0.5 | 75.4 ± 0.3 |
| ST-ProtoPNet | **81.8 ± 0.3** | 82.9 ± 0.4 |
| TesNet | 80.9 ± 0.2 | 82.1 ± 0.3 |

Table 3: Classification accuracy (ACC) on ImageNet dataset by competing approaches using different CNN backbones.

| CNN Model | ACC [%] | Transformer Model | ACC [%] |
| --- | --- | --- | --- |
| ResNet-34 | 73.3 | ViT-B/16 | 81.1 |
| ↳InfoDisent | 64.1 ± 0.39 | ↳InfoDisent | 79.2 ± 0.21 |
| ResNet-50 | 76.1 | Swin-S | 83.4 |
| ↳InfoDisent | 67.8 ± 0.05 | ↳InfoDisent | 81.4 ± 0.06 |
| DenseNet-121 | 74.4 | MaxVit | 83.4 |
| ↳InfoDisent | 66.6 ± 0.02 | ↳InfoDisent | 83.3 ± 0.11 |
| ConvNeXt-L | 84.1 | | |
| ↳InfoDisent | 82.8 ± 0.09 | | |

**Does InfoDisent reduce channel correlation** To evaluate the effectiveness of our method in reducing channel correlation, we employed the RV coefficient Robert & Escoufier (1976), a standard metric for measuring linear dependence between sets of variables. As shown in Tab. 4, the results across 14 experimental configurations (spanning 3 datasets and 7 backbone architectures) reveal that our model achieves lower (i.e., better) RV coefficients than the baseline in 9 cases, performs comparably in 3 cases, and underperforms in only 2 cases. Notably, although our approach does not include an explicit loss term designed to minimize the RV coefficient, it consistently demonstrates superior or equivalent capability in reducing channel correlations compared to the baseline.

Table 4: RV correlation between channels for InfoDisent and baseline models.

| Model | Cars | | CUB | | ImageNet | |
|---|---|---|---|---|---|---|
| | InfoDisent | Baseline | InfoDisent | Baseline | InfoDisent | Baseline |
| ConvNeXt-L | – | – | – | – | 8.7 | **3.0** |
| ConvNeXt-Tiny | **6.7** | 6.9 | **5.5** | **5.5** | – | – |
| DenseNet121 | **6.2** | 6.9 | **7.2** | 7.9 | 8.4 | **4.5** |
| ResNet34 | **6.0** | 7.1 | **6.5** | 7.2 | **5.2** | 6.0 |
| ResNet50 | **6.2** | 7.6 | **4.9** | 7.8 | **3.0** | 6.5 |
| Swin-S | – | – | – | – | **2.5** | **2.5** |
| ViT-B/16 | – | – | – | – | **9.9** | **9.9** |

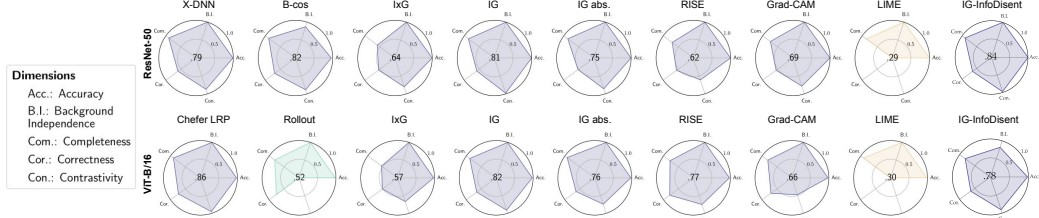

Figure 7: FunnyBirds evaluation results for various XAI methods. Model-agnostic methods are assessed on ResNet-50 and ViT-B/16. Results are averaged over the entire test set, including the center score representing the mean of completeness (Com.), correctness (Cor.), and contrastivity (Con.) dimensions. Additionally, accuracy (Acc.) and background independence (B.I.) are reported. Our approach (at the end on the right) enhances model explainability, showing significant improvements for ResNet-50 and satisfactory results for the transformer model. In the Appendix Tab. 9 are detailed results.

## 5 EVALUATION OF INTERPRETABILITY

We showcase the performance of InfoDisent on explainability metrics using FunnyBirds dataset Hesse et al. (2023) and Spatial Misalignment (SM) Sacha et al. (2024) which are dedicated benchmarks for XAI methods. Results for SM are presented in the Appendix Tab. 7. Also, we performed tests related to diversity of the concepts Tab. 15, prototype robustness Tab. 10, and number of utilized concepts in Appendix Tab. 16. In addition to the computational validation of InfoDisent, we conducted two user studies. The first study followed the design of the HIVE benchmark Kim et al. (2022), aiming to assess the impact of explanations on user overconfidence in the model's predictions. The second study evaluated the disambiguation of explanations derived from prototypical parts. For this, we adopted a study design from prior works on prototypical parts Ma et al. (2023); Pach et al. (2024).

**FunnyBirds results.** To evaluate explainability of InfoDisent approach, we utilized the Funny-Birds Hesse et al. (2023) dataset, which is constructed to evaluate the semantic appropriateness of explanations through semantically meaningful image interventions, such as removing individual object parts. This enables a more nuanced analysis of explanations at the part level, which aligns more closely with human understanding compared to pixel-level evaluations. We compare InfoDisent approach to a range of XAI methods for two different backbones Fig. 7. InfoDisent enhances the explainability of models based on classic CNN architectures and ranks best when using the ResNet50 backbone. While still highly competitive, it achieves the third-best result for the ViT.

### 5.1 USER STUDY RESULTS

We performed two user studies. Each of them involved 60 participants per dataset, with a balanced gender representation. Participants were aged between 18 and 60, with an average age of 35 years. The studies were conducted on the Clickworker platform, using two datasets: CUB-200-2011 and ImageNet. Each participant answered 20 questions, with images randomly selected from the testing dataset for each question. Example questions are provided in the Supplementary Materials.

**Confidence in Model Predictions** In the first study, which assessed user overconfidence, participants were presented with an image alongside the model's explanation. They were then asked, "What do you think about the model's prediction?" and were instructed to indicate their confidence in whether the model was correct or incorrect as it was done in HIVE benchmark Kim et al. (2022).

The results of this user study are shown in Tab. 5. They demonstrate that users evaluating the model's predictions based on explanations from InfoDisent are statistically significantly less overconfident than random guessing (as indicated by the p-values). Additionally, InfoDisent generalizes to ImageNet and achieves statistically significantly better results than random.

**Disambiguity of prototypical parts.** In the second study, which assessed the disambiguation of prototypical parts, participants were shown an image classified by the model alongside two explanations corresponding to the two most activated classes. Their task was to determine, based on the explanations, which decision the model had made.

The results of this user study, presented in Tab. 6, demonstrate that users who saw explanations from InfoDisent performed statistically significantly better than random guessing.

For both of the user study we present results from other user groups in the Appendix Tabs. 13 and 14 just for a reference.

Table 5: Results showcasing user confidence in model predictions reveal that when users answered questions informed by explanations from our InfoDisent model, they exhibited substantial confidence in the model's correct decisions across both datasets (mean confidence ober 60% for ImageNet and over 80% for CUB). However, a notable challenge emerged: users struggled to detected samples where the model's predictions were incorrect, even when provided with the explanations. This finding echoes observations reported for other XAI methods. We denote statistically significant values in bold.

| Method | Prediction | ImageNet [%] | CUB-200 [%] |
|---|---|---|---|
| | Correct | $60.2 \pm 9.0$ | $80.7 \pm 13.3$ |
| InfoDisent | Incorrect | $55.3 \pm 9.9$ | $42.7 \pm 11.7$ |
| | p-value | $0.001$ | $3 \cdot 10^{-15}$ |

Table 6: A user study assessing the perceived ambiguity of prototypical parts shows that explanations provided by InfoDisent allows users to understand the model's decisions significantly better than random guessing on both ImageNet and CUB ($p < 0.05$). Notably, InfoDisent is the only method leveraging prototypical parts that operates on the ImageNet. Furthermore, users interacting with InfoDisent achieved a level of understanding that is statistically significant We report the p-value associated with the comparison against random guessing.

| Method | Dataset | User Acc. [%] | p-value |
|---|---|---|---|
| InfoDisent | ImageNet | $59.3 \pm 14.9$ | $8 \cdot 10^{-6}$ |
| | CUB-200 | $64.7 \pm 13.1$ | $10^{-14}$ |

## 6 CONCLUSIONS

In this work, we introduce InfoDisent, an innovative model that combines the strengths of both post-hoc and inherently interpretable methods. InfoDisent provides the flexibility to be applied to any backbone, while offering both local (per image) and global (per class) explanations in the form of atomic concepts, addressing key limitations of existing approaches. Additionally, as demonstrated by user studies, InfoDisent performs comparably to state-of-the-art methods in disambiguating prototypical parts and managing user overconfidence. Notably, InfoDisent is the first attempt to generalize the prototypical parts-based methodology to big scale datasets such as the whole ImageNet. In future work, we plan to explore pruning techniques to optimize the size of concepts used in explanations.

**Limitations.** While achieving interpretability, InfoDisent requires training its classification head on the complete dataset. Unlike prototypical parts models, the model's decisions are not constrained to a fixed number of prototypes. Furthermore, because InfoDisent avoids backbone fine-tuning, the quality of its explanations is inherently limited by the expressiveness of the initial backbone representations. Moreover, all prototypical-parts based model may have generate concepts that are not human understandable, which is also a limitation of our InfoDisent model.

**Ethics statement.** InfoDisent advances the field of Explainable AI (XAI) by introducing a novel method that generalizes prototypical parts-based explanations to ImageNet-like datasets, while maintaining the post-hoc flexibility of application. InfoDisent holds potential for further exploration in downstream applications, such as medical diagnosis.

**Reproducibility Statement.** We make the code available through zip file in the submission system, and we plan to put public github into the camera ready version. Moreover, we ran experiments on NVIDIA RTX4090 and NVIDIA A100 40GB. Also, we provided all details regarding the hyperparameters to reproduce the results in the Appendix Section C.

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

# SUPPLEMENTARY MATERIAL

## A    ABLATION STUDY

InfoDisent organizes information into channels with sparse representations, which can be later utilized in the model's prediction process. In the following experiment, we investigate how the number of channels affects the model's predictions. Specifically, we assess how many channels are required to account for at least 95% of the information used in the model's predictions. Formally, if logits $= \sum_{i=0}^{N} a_{ki} v_i + b_k$, where $N$ is a number of all channels, $k$ represents the image class, and $a_{ki}, v_i, b_k \in \mathbb{R}$, then for each image from class $k$, we determine the smallest number $n$ channels such that $\sum_{i \in I_k} |a_{ki} v_i| / \sum_{i=0}^{N} |a_{ki} v_i| \geq 0.95$, where $I_k$ is the set of indexes of the $n \leq N$ largest values of $|a_{ki} v_i|$. This analysis allows us to identify the most critical channels contributing to the model's decisions, providing deeper insights into the model's interpretability and efficiency.

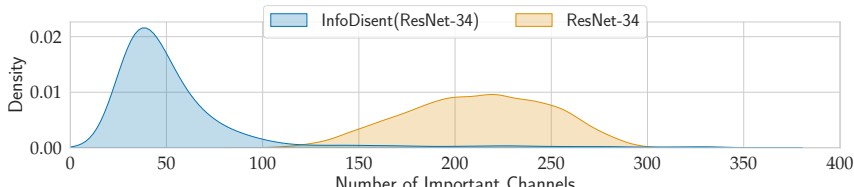

Figure 8: Density estimate of the number of significant channels for each class and image in the CUB-200-2011 test set using the ResNet-34 network. InfoDisent uses significantly fewer channels and has less variance.

Fig. 8 shows the density estimate of the number of significant channels for each class and image in the CUB-200-2011 test set. In this experiment, we used the ResNet-34 network that utilizes a significantly larger number of channels in its predictions compared to InfoDisent model. By reducing the number of significant channels while preserving classification performance, our model demonstrates efficient resource utilization and improved interpretability, as illustrated below. This efficiency shows that our model is more effective at isolating the critical features necessary for accurate predictions, thereby validating our approach. This also validates the disentangling role of the orthogonal matrix $U$.

**Optimization of U and W**   Our objective is to represent a matrix $A \in \mathbb{R}^{n \times m}$ as a product of matrices $A \approx SU$, where $S$ is sparse and $U$ is unitary. Given the equality $A = SU$, we would have $S = AU^{-1}$. To obtain the sparsest possible $S$, we can formulate the minimization problem:

$$\min_{U \text{ unitary}} \|AU^{-1}\|_1.$$

We leverage the following key properties:

- For any square matrix $B \in \mathbb{R}^{n \times n}$, the matrix $\frac{1}{2}(B - B^T)$ is skew-symmetric.
- If $B$ is skew-symmetric, then its matrix exponential $\exp(W)$ is orthogonal (and consequently unitary for real matrices).

This leads to the reformulated optimization problem:

$$\min_{B \text{ skew-symmetric}} \|A \exp(-B)\|_1,$$

where our solution becomes $S = A \exp(-B)$.

The final unitary matrix is computed as:

$$U = \exp\left(-\frac{1}{2}(W - W^T)\right),$$

where:

- $W$ is a randomly initialized matrix
- The optimization is performed using gradient descent
- We employ the Adam optimizer for efficient convergence

This parametrization guarantees that $U$ remains unitary throughout the optimization process while allowing efficient computation of gradients through the matrix exponential operation.

**Additional definitions** Here we present a more precise, quantitative definition of "prominently" and "consistency" in our channel selection process. For each test image $x_t$ from the test set $X_{test}$, we first identify the top 10 most activated channels. Let $A(C_i, x_t)$ denote the activation value of channel $C_i$ for image $x_t$. We select the set of 10 channels, $K_t$, such that for any $C_j \in K_t$ and $C_k \notin K_t$, $A(C_j, x_t) \geq A(C_k, x_t)$ after sorting activations in descending order.

To define "prominently" and "consistency" for a given class $Y$, we then count how many times each channel $C_i$ appears within the top 10 activated channels across all test images belonging to that class. Let $N(C_i, Y)$ be this count:

$$N(C_i, Y) = \sum_{x_t \in X_{test}, \text{label}(x_t) = Y} \mathbf{1}[C_i \text{ is among the top 10 activated channels for } x_t]$$

A channel $C_i$ is then selected for visualization for class $Y$ if its count $N(C_i, Y)$ exceeds a predefined threshold. Specifically, we select channels where:

$$N(C_i, Y) > 0.50 \times |\{x_t \in X_{test} \mid \text{label}(x_t) = Y\}|$$

This procedure ensures that only channels that are consistently among the most activated (i.e., prominently contributing) across a significant portion (more than 50%) of the test images for a given class are selected for visualization.

# B  DETAILS OF THE EXPERIMENTS PERFORMED

**Datasets** In our experiments, we leveraged several diverse datasets to evaluate performance. The first dataset is the Caltech-UCSD Birds-200-2011 (CUB-200-2011)Wah et al. (2011), which contains 11,788 images meticulously labeled across 200 bird species, divided into 200 subcategories. Of these, 5,994 images are allocated for training, while 5,794 are reserved for testing. The second dataset, known as Stanford CarsKrause et al. (2013), is designed to classify various car models. It includes 16,185 images, each capturing a rear view of different car types across 196 classes, with an almost even distribution between training (8,144 images) and testing (8,041 images) subsets. Each class details the car's make, model, and year. The third dataset, Stanford Dogs Khosla et al. (2011), features a collection of 20,580 images representing 120 dog breeds from around the globe. This dataset, sourced and annotated through ImageNet, is intended for fine-grained image classification, with 12,000 images for training and the remainder for testing.

Additionally, we incorporated the FunnyBirds Hesse et al. (2023) dataset, consisting of 50,500 images representing 50 synthetic bird species, with 50,000 images for training and 500 for testing. This dataset was designed with a focus on "concepts," or mental representations crucial for categorization, and is particularly relevant for explainable AI (XAI). The concepts are linked to specific bird anatomy parts, such as the beak, wings, feet, eyes, and tail, ensuring they are both granular and intuitive for practical use in XAI.

Finally, we utilized ImageNet Russakovsky et al. (2015), a highly recognized dataset in computer vision, often employed for pretraining deep learning models. ImageNet encompasses 1,281,167 training images, 50,000 validation images, and 100,000 test images, spanning 1,000 object classes.

**Training Details** We train the architectures using stochastic gradient descent (SGD) with standard categorical cross-entropy loss. The momentum, damping, and weight decay are set to 0.9, 0.9, and 0.001, respectively. For the baseline networks, the initial learning rates are 0.1, 0.05, and 0.01, which are reduced by a factor of 0.1 when the validation loss converges. In our approach, we train only

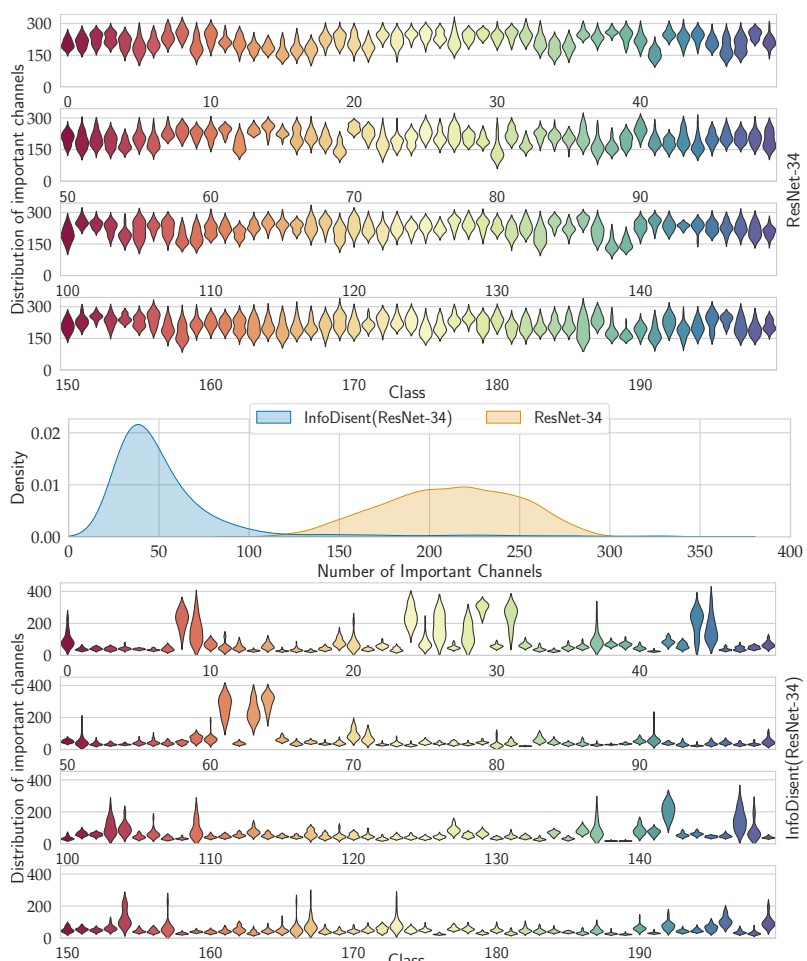

Figure 9: Density estimate of the number of significant channels for each class and image in the CUB-200-2011 test set using the ResNet-34 network. InfoDisent uses significantly fewer channels, particularly evident in the middle image, which shows the density estimation of the number of significant channels for all classes.

the last two segments of the network, thus we use lower learning rates of 0.001 and 0.0001, utilizing the 'ReduceLROnPlateau' Al-Kababji et al. (2022) mechanism that reduces the learning rate when the cost function stops improving. All numerical experiments were conducted using NVIDIA RTX 4090 and NVIDIA A100 40 GB graphics cards.

For cropped images, we follow previous studies Chen et al. (2019) by applying on-the-fly data augmentations (e.g., random rotation, skew, shear, and left-right flip) on the cropped CUB and cropped Cars datasets using the provided bounding boxes. We also validate our method on the full (uncropped) CUB and Dogs datasets, employing the same online data augmentation techniques (e.g., random affine transformation and left-right flip). For the FunnyBirds dataset, we adhered to the detailed instructions provided in the framework's documentation, which can be found at `https://github.com/visinf/funnybirds`. For training various CNN and transformer models on the ImageNet dataset, we utilized the augmentation techniques described at `https://github.com/pytorch/vision/tree/main/references/classification`.

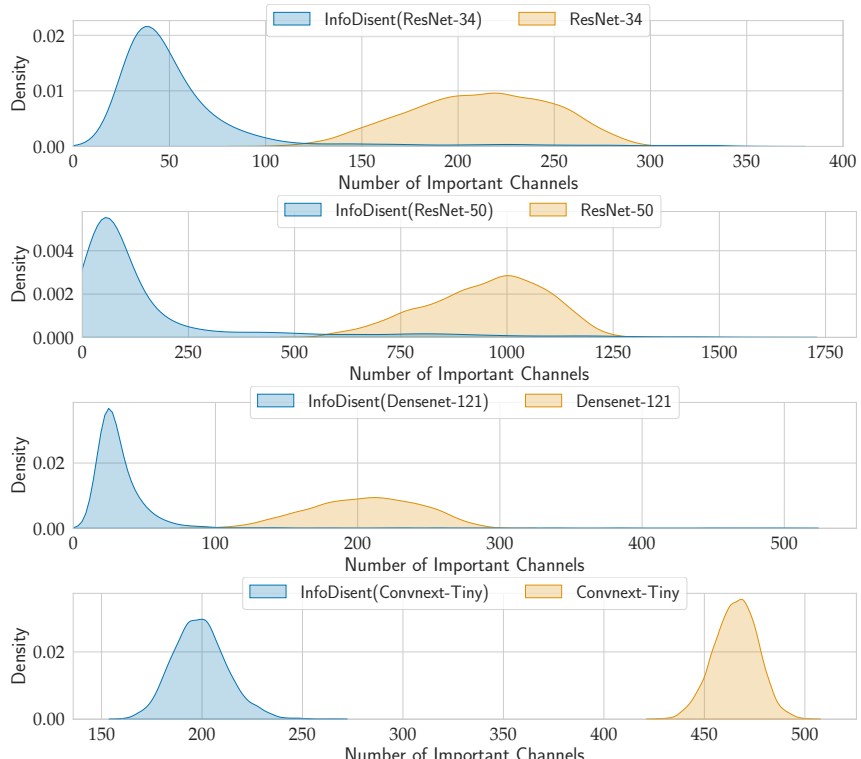

Figure 10: Density estimate of the number of significant channels for each class and image in the CUB-200-2011 test set using various networks.

## C ADDITIONAL RESULTS

In this section, we provide additional results to supplement and expand upon the findings presented in the main part of the paper. We delve deeper into the behavior of our model across various experiments and datasets, offering a more comprehensive analysis.

**Spatial Misalignement** Our method, InfoDisent, trained on the CUB-200-2011 dataset with a DeiT backbone, achieves the best scores in two of the three metrics on this benchmark. However, it is more susceptible to a loss in accuracy. This may be related to the difference in backbones; other methods fine-tune their backbones, whereas InfoDisent does not.

Table 7: Results on challenging Spatial Misalignement Benchmark. Note that InfoDisent scores best in 2 out of 3 interpretability metrics.

| Model | PAC | PLC | PRC | Acc. Before | Acc. After | AC |
|---|---|---|---|---|---|---|
| InfoDisent | **0.11** | **0.37** | 13.66 | 83.66 | 67.21 | 16.45 |
| ProtoViT | 2.92 | 21.68 | **1.28** | 85.40 | 82.80 | **2.60** |
| ProtoPNet | 23.70 | 24.00 | 13.50 | 76.40 | 68.20 | 8.20 |
| TesNet | 3.40 | 16.00 | 2.90 | 81.60 | 75.80 | 5.80 |
| ProtoPool | 11.20 | 31.80 | 4.50 | 80.80 | 76.00 | 4.80 |
| ProtoTree | 23.70 | 27.70 | 13.50 | 76.40 | 68.20 | 8.20 |

**Performance gains** We measured and compared the following: (1)Training time per epoch for our InfoDisent model with a frozen backbone versus a full version of the model. (2) The number of parameters being optimized in each scenario. (3) The GPU memory usage.

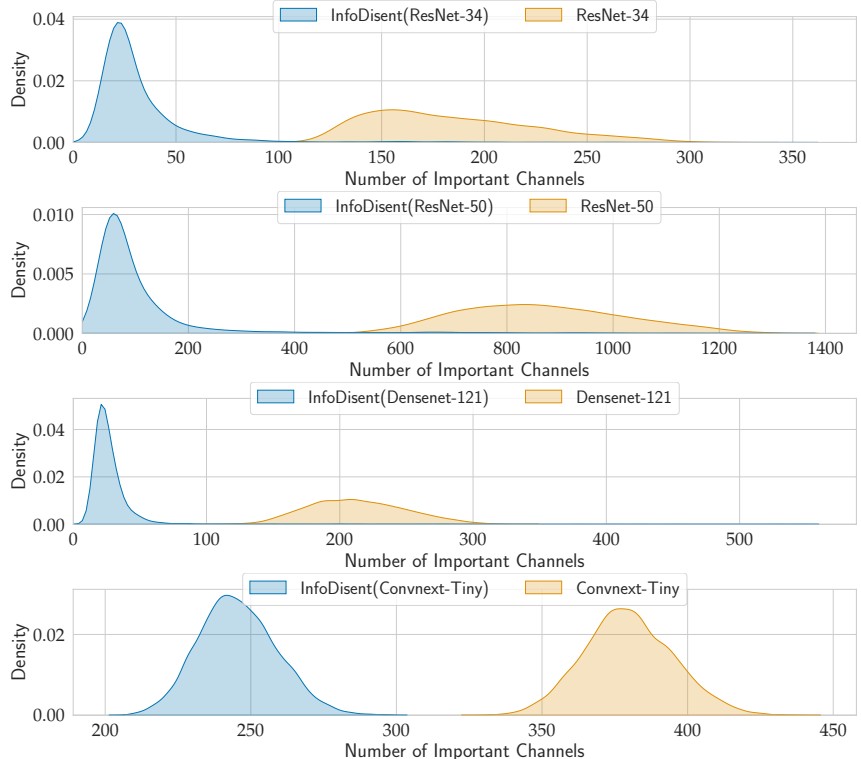

Figure 11: Density estimate of the number of significant channels for each class and image in the Stanford Cars test set using various networks.

As can be observed in the table below, with InfoDisent we need to spend 3 times less time on average per training epoch, significantly less parameters needs to be optimized and we need up to 75% less GPU memory.

The averaged results from these experiments will be included in the manuscript.

Note that, to ensure a fair evaluation, we used the same hardware (NVIDIA A100), dataset (CUB), and batch size.

Table 8: Model Training and Resource Consumption Comparison.

| Model | Time p. Epoch [s] | No. Param. | Mem. GPU [GB] |
|---|---|---|---|
| ConvNeXt-Tiny (Full) | 664 | 28,563,752 | 5.84 |
| ConvNeXt-Tiny (InfoD) | 205 | 745,160 | 1.85 |
| ResNet50 (Full) | 652 | 28,112,136 | 5.26 |
| ResNet50 (InfoD) | 498 | 4,604,104 | 2.67 |
| Swin-S (Full) | 1500 | 49,712,066 | 11.16 |
| Swin-S (InfoD) | 480 | 743,624 | 2.37 |

**Ablation study**   In this part, we present additional results from a series of analyses investigating the significance of the number of channels on model predictions across various datasets and models. The results of these analyses are illustrated in Figs. 9 to 12. Recall, that InfoDisent model organizes information into channels with sparse representations, which are later utilized in the model's prediction process. We specifically examine how the number of channels influences the model's predictions by determining the minimum number of channels required to account for at least 95% of the information used in the model's predictions.

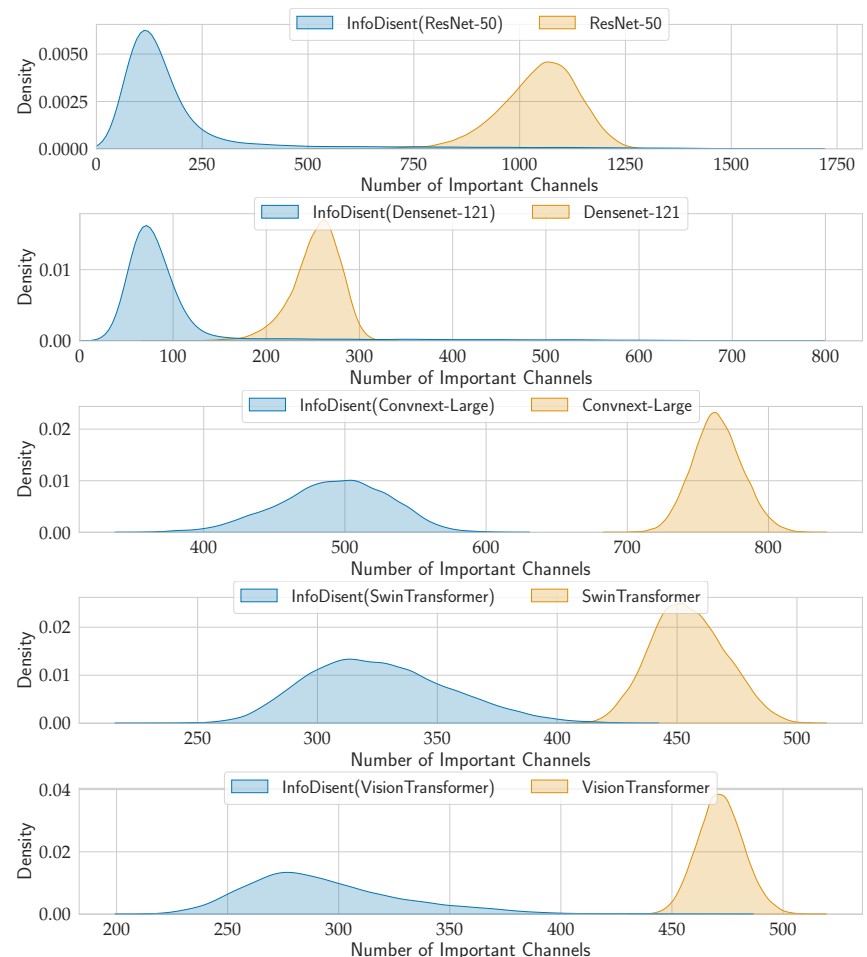

Figure 12: Density estimate of the number of significant channels for each class and image in the ImageNet test set using various networks.

Formally, if

$$\text{logits} = \sum_{i=0}^{N} a_{ki} v_i + b_k,$$

where $N$ is the total number of channels, $k$ represents the image class, and $a_{ki}, v_i, b_k \in \mathbb{R}$, then for each image from class $k$, we identify the smallest number $n$ of channels such that

$$\frac{\sum_{i \in I_k} |a_{ki} v_i|}{\sum_{i=0}^{N} |a_{ki} v_i|} \geq 0.95,$$

where $I_k$ is the set of indexes of the $n \leq N$ largest values of $|a_{ki} v_i|$.

This analysis highlights the most critical channels contributing to the model's decisions, providing deeper insights into the model's interpretability and efficiency. Note that the InfoDisent approach consistently utilizes significantly fewer channels, a trend observed across all models and datasets analyzed.

Figs. 13 and 14 also present the channel values before the final linear layer in our model for randomly selected images from various datasets and models. As evident from the images, our model utilizes a significantly smaller number of channels in its predictions compared to the baseline models.

**Explaining Classification Decision for a Given Image by Prototypes** InfoDisent employs prototypes, similar to the approach used in PiPNet Nauta et al. (2023a), to explain individual decisions

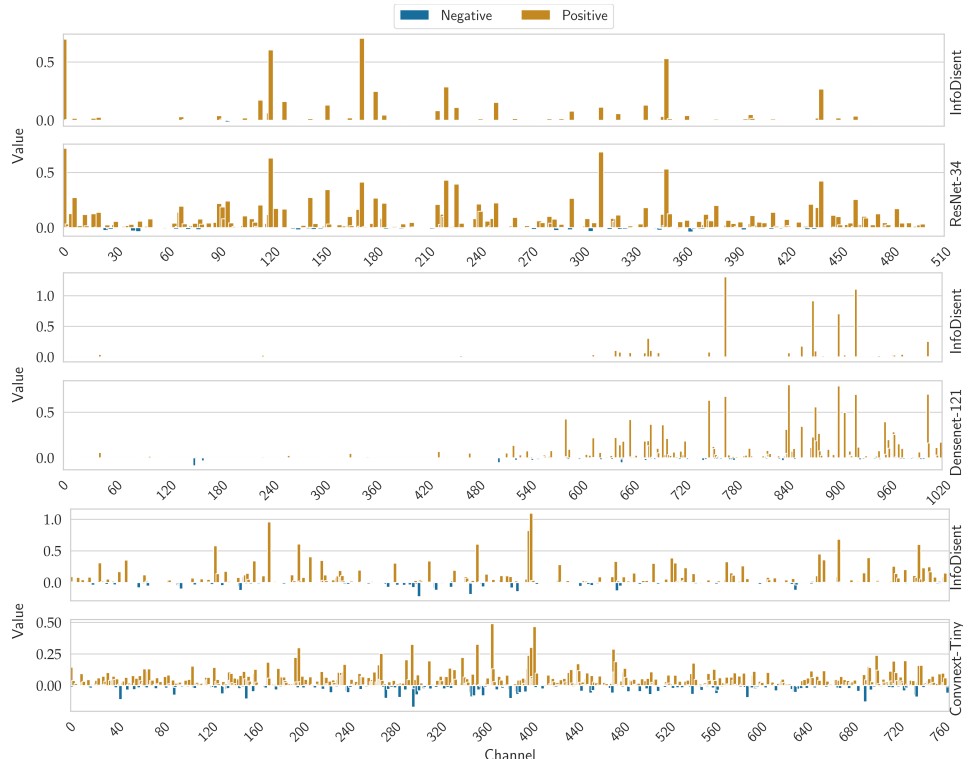

Figure 13: Channel activations before the final linear layer for a randomly selected image processed by different models trained on the CUB-200-2011 dataset are shown. The results are displayed in three groups, each containing two graphs. Model names are listed on the right side of the graphs. Unlike the baseline models, our network activates significantly fewer channels while still maintaining strong performance.

made on an image. Below, we describe the process for identifying prototypes for a given input image. An average pooling operation makes the aggregation of information from individual channels. The outcome of this operation is a scalar for each channel $K$, computed as

$$\mathrm{mx\_pool}(K) = \max(\mathrm{ReLU}(K)) - \max(\mathrm{ReLU}(-K))$$

(as explained in the main paper). This dense representation, formed by aggregating all the channels, is then processed by a linear layer that outputs logits, see Fig. 2. The logits can be represented in a format similar to the output of a convolutional layer, as illustrated by $V_+$ and $V_-$ in Fig. 2. This approach maintains the pictorial structure of the logits, allowing us to extract individual channels.

Note that each channel in InfoDisent model can contain only two possible values (refer to Fig. 4a in the main paper, where these values are depicted as red or blue areas within the channels). To identify the prototype, i.e., the relevant channel, we focus solely on the positive values within the channels (represented by red areas in Fig. 4a). These positive values indicate the significant part of the channel/prototype (marked by the yellow frame on the prototype) and define the channel's importance for the model prediction (since the linear layer uses only nonnegative coefficients). As demonstrated in Figs. 13 and 14, the number of such channels is limited (alternatively, we could also focus on channels with the strongest values). Once we have identified the important channels (by knowing their indices), we represent each channel using prototypes. Prototypes are images from the training set that exhibit the five strongest positive values for a given channel. Example results illustrating how model predictions are explained using prototypes are shown in Figs. 15 to 19 (which are after the references).

Figs. 15 to 17 showcase the performance of our prototype models on standard benchmarks: CUB-200-2011, Stanford Cars, and Stanford Dogs. Fig. 15 demonstrates the model's ability to focus on distinctive features like the Scissor-tailed Flycatcher's elongated tail feathers, underwing yellow coloration, or the Red-legged Kittiwake's red feet.

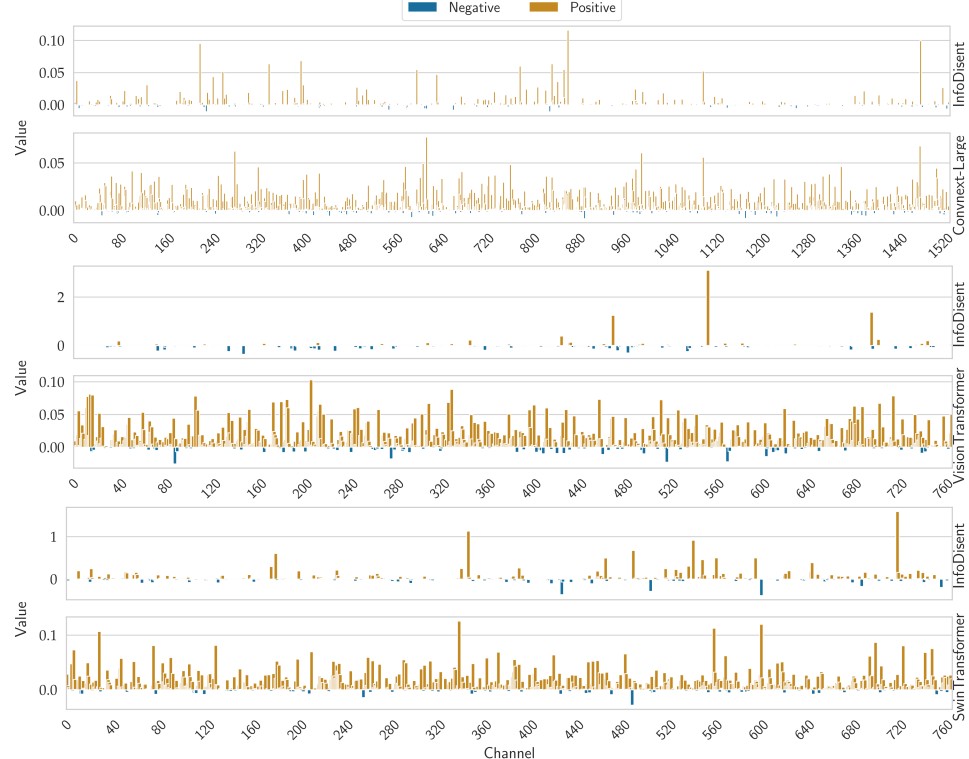

Figure 14: Channel activations before the final linear layer for a random image processed through various ConvNeXt, Vision Transformer, and Swin Transformer models trained on ImageNet. The results are displayed in three groups, each containing two graphs. Model names are listed on the right side of the graphs. InfoDisent activates significantly fewer channels compared to the baseline models while maintaining high effectiveness.

Similarly, Fig. 16 highlights the model's capacity to identify key vehicle components. For instance, it accurately pinpoints the fender, bumper, and body of a Jeep Wrangler SUV 2012, and the characteristic stripes on a Ford GT Coupe 2006.

Complementing our experiments with widely used CNN models, we investigated the performance of transformer architectures, specifically VisionTransformer (ViT-B/16) Dosovitskiy et al. (2020), SwinTransformer (Swin-S) Liu et al. (2022a). Their results are visualized in Figs. 18 and 19. Our analysis reveals that transformer models concentrate on smaller image regions than CNNs. Nevertheless, both model types generate interpretable prototypes that offer insights into the input image content.

**Decision Behind Class**    To delve deeper into the model's decision-making process, we employ a prototype-based analysis at the class level. For each class, we identify key channels that consistently exhibit strong activation across the entire test set. These channels, indicative of the model's focus on specific visual features, are selected based on their prominence and reliability in representing the class. By visualizing these key channels as prototypes, as demonstrated in our previous image analysis, we obtain a clear representation of the model's class-specific decision criteria.

Figs. 20 to 22 present the results of our prototype analysis for selected classes in the CUB-200-2011 and Stanford Cars datasets, using both InfoDisent(ResNet-50) and transformer models. This visualization allows for a granular understanding of how these models differentiate between various classes, highlighting the underlying patterns recognized by the network.

**Heatmaps**    Figs. 23 and 24 present example heatmaps generated by our model, resembling those produced by Grad-CAM Selvaraju et al. (2017). Our method, rooted in representational channels, simplifies heatmap generation by accumulating activations from all prototypes (positive and nega-

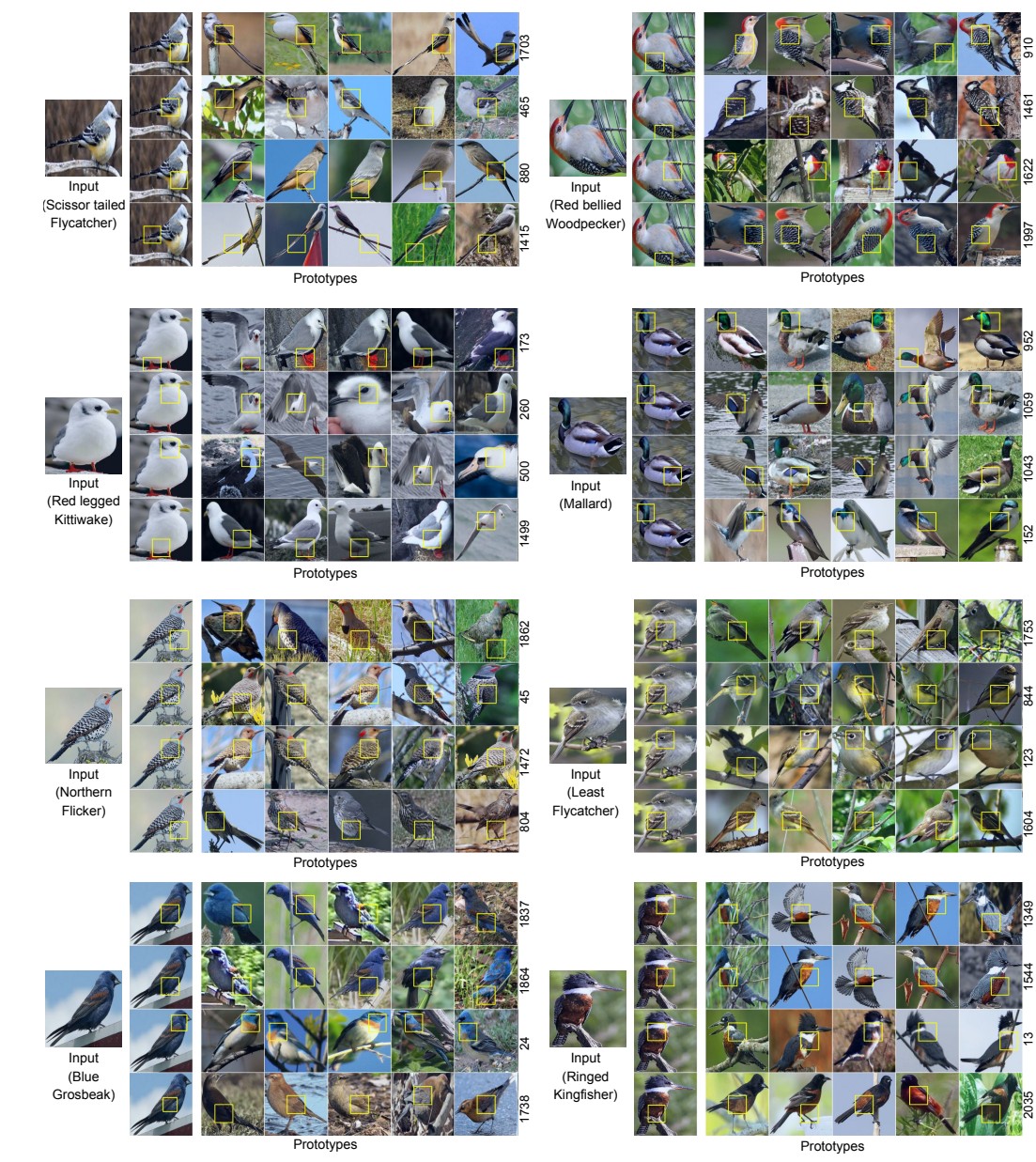

Figure 15: Example prototypes generated by InfoDisent(ResNet-50) models for random input images from the test set of the CUB-200-2011 dataset. The display includes 8 input images, each with a corresponding column where yellow boxes highlight specific regions, followed by the prototypes (images on the right). Each row represents prototypes from a different channel, with the channel index on the right. Observe that the prototypes identified by the model effectively capture distinct parts of the body in the images.

tive) across all channels. Leveraging the information bottleneck principle, our approach yields more focused heatmaps compared to traditional methods.

**Detailed results on FunnyBirds** In Tab. 9, we present details regarding the results on FunnyBirds framework.

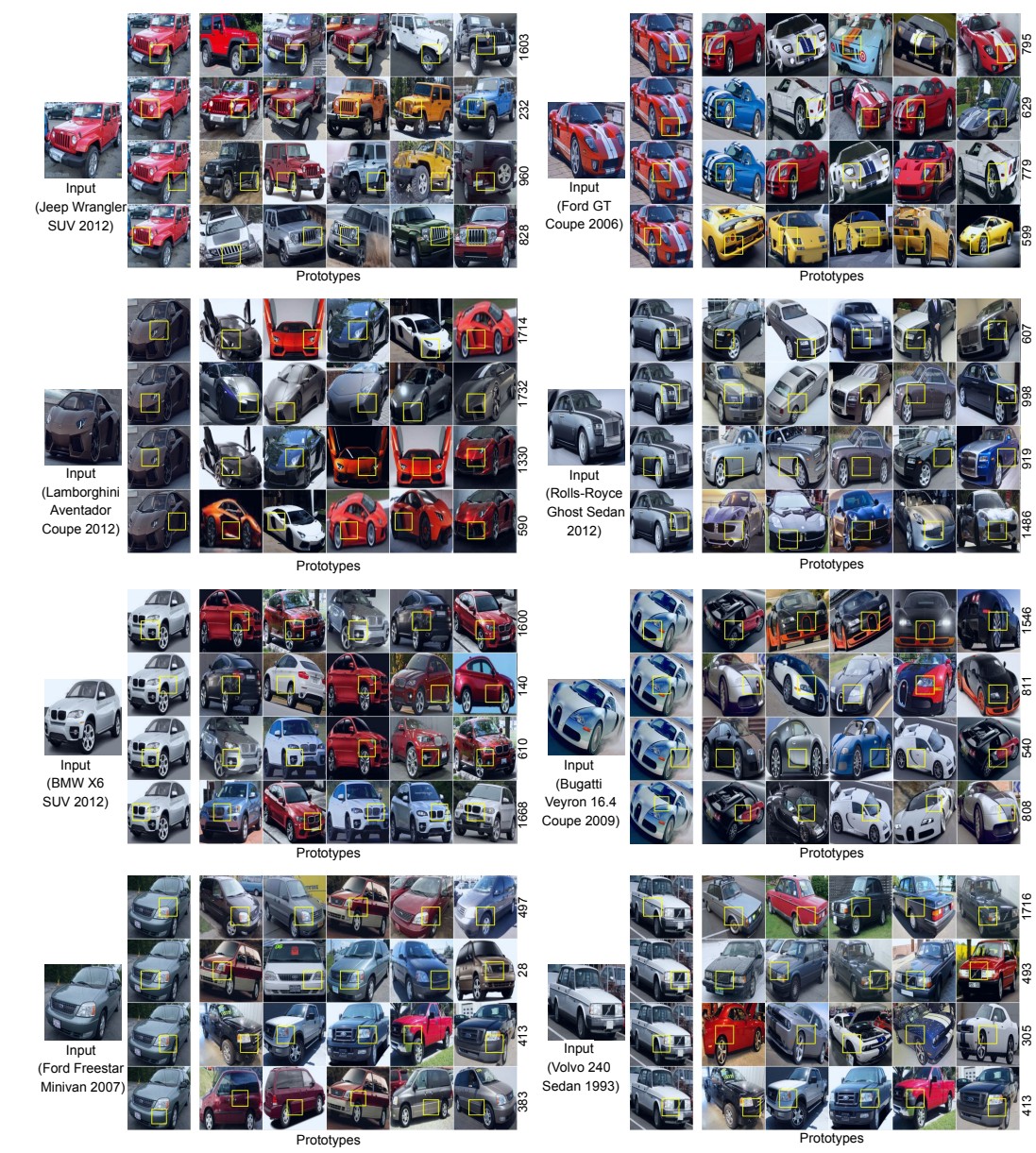

Figure 16: To demonstrate InfoDisent(ResNet-50) model's capability, we visualize prototypes generated for arbitrary test images from the Stanford Cars dataset. Each image is paired with highlighted areas and its associated prototypes, grouped by channel. The results indicate that the model has learned to extract meaningful features representing different vehicle parts.

**Prototypical parts robustness**  We evaluated stability against Gaussian noise and transformations (brightness, contrast, saturation) as in ProtoPShare Rymarczyk et al. (2021) to measure the robustness of prototypical parts.

For the evaluation, we examined how InfoDisent's prototypes adapted to perturbed inputs using two similarity measures: Jaccard Similarity Index (J). Cosine Similarity (C).

InfoDisent consistently demonstrated strong performance across these metrics.

**More details on user study**  Each worker was paid €2.00 for completing a short 20-question survey. The survey questions were randomly composed, so the specific questions differed between

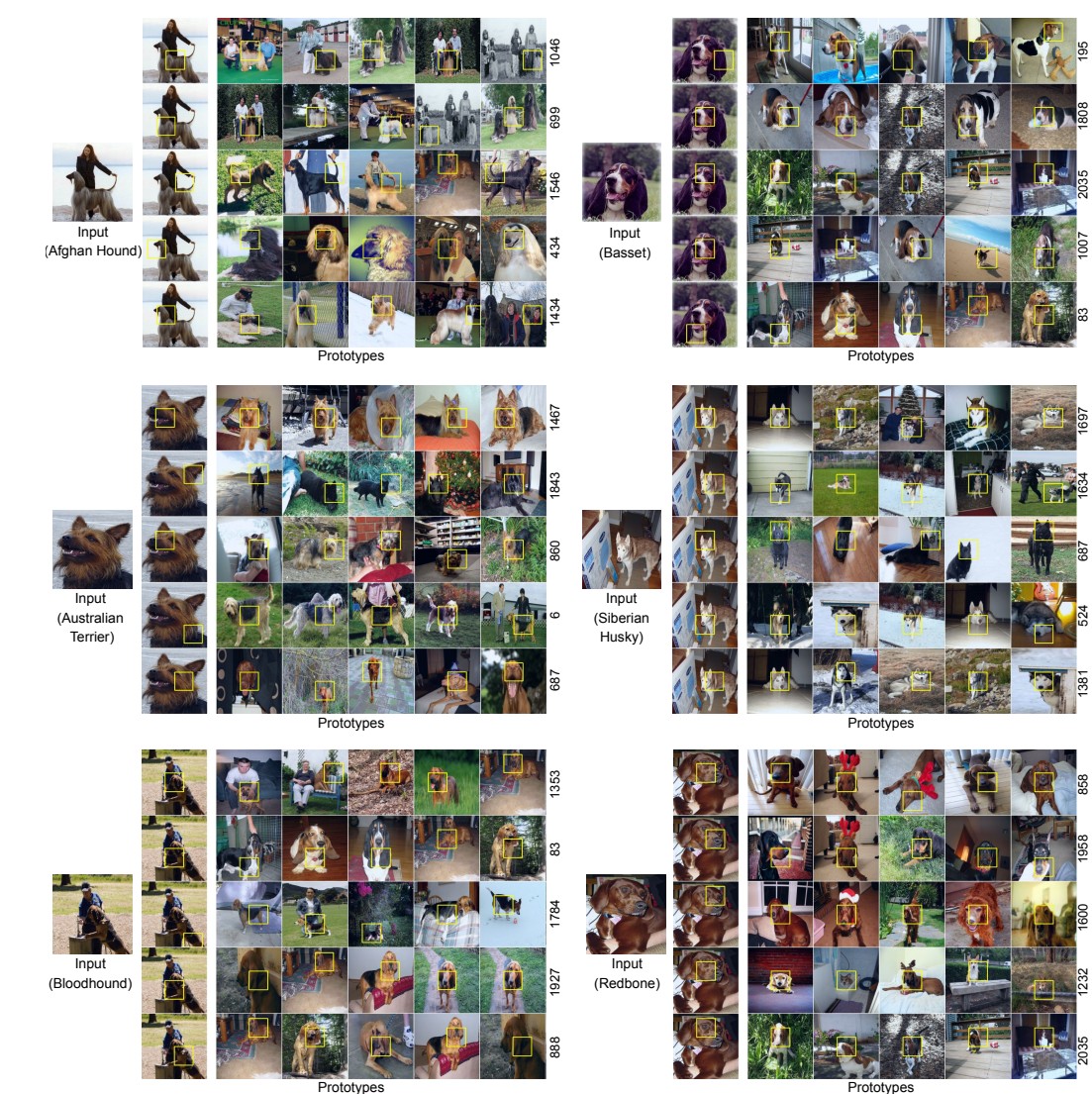

Figure 17: Visualized prototypes from the Stanford Dogs dataset. Each row highlights prototypes from a specific channel, focusing on different dog features such as ears, nose, and fur.

participants. The participants were gender-balanced and ranged in age from 18 to 60. They were given 30 minutes to complete the survey.

To ensure data quality, we excluded responses where users selected the same answer for all questions. Surveys were repeated until we obtained 60 valid responses. Figs. 25 and 26 illustrate example questions used in both user studies.

Before starting the survey, participants were provided with an example and detailed instructions to familiarize them with the study setup, including the explanation composition and visualization. The distribution of answers is summarized in Tabs. 11 and 12.

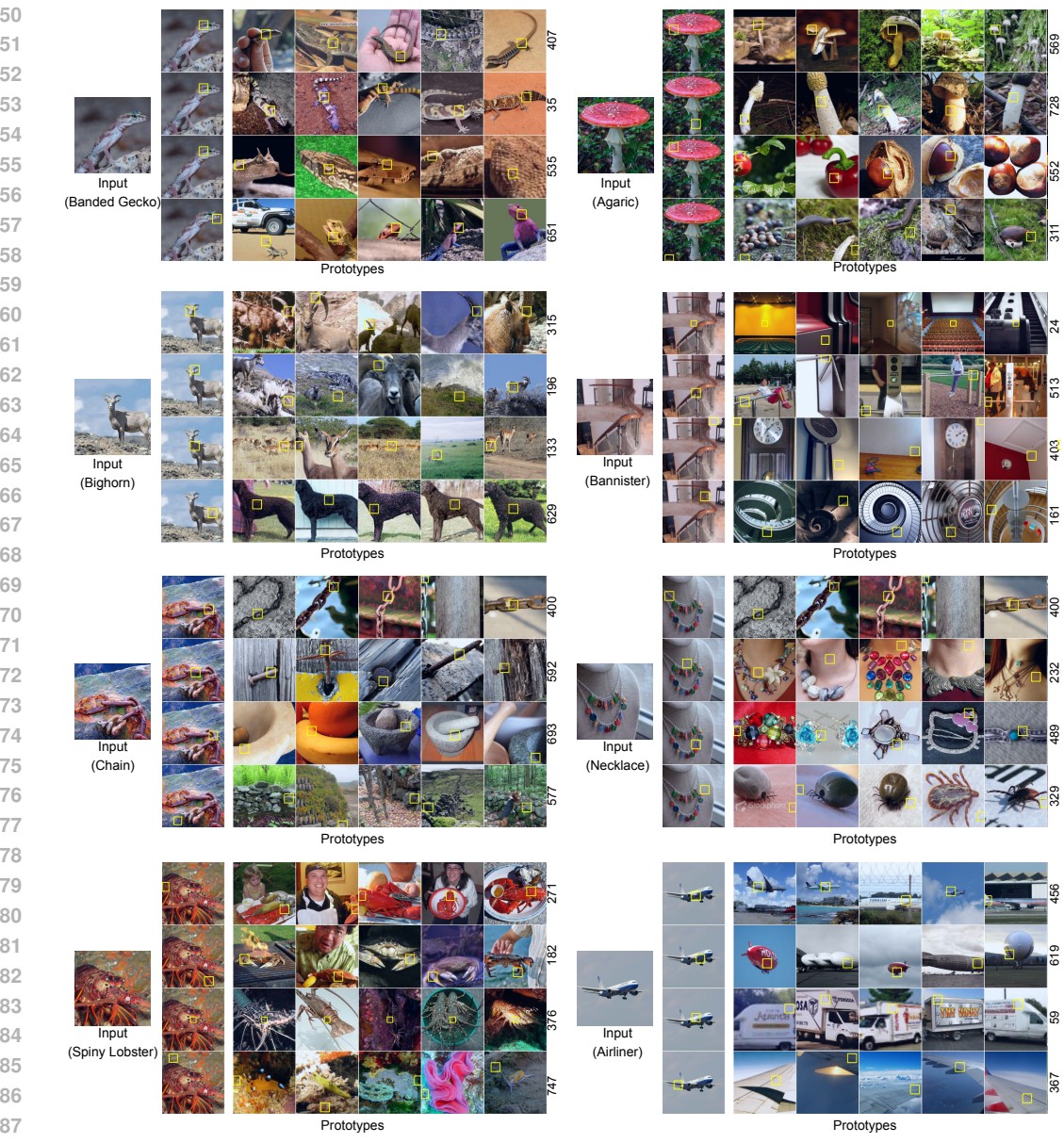

Figure 18: This figure showcases the prototypical explanations provided by InfoDisent(ViT-B/16) model. We visualize prototypes for arbitrary ImageNet images, highlighting relevant regions and their corresponding channel-specific prototypes.

Table 13: Results showcasing user confidence in model predictions. We denote statistically significant values in bold. Results for ProtoPNet and GradCAM, marked with an asterisk (*), are sourced from the original HIVE study Kim et al. (2022).

| Method | Prediction | ImageNet [%] | CUB-200 [%] |
|---|---|---|---|
| ProtoPNet* | Correct | NA | $73.2 \pm 24.9$ |
| | Incorrect | NA | $\mathbf{46.4 \pm 35.9}$ |
| GradCAM* | Correct | $70.8 \pm 26.6$ | $72.4 \pm 21.5$ |
| | Incorrect | $\mathbf{44.8 \pm 31.6}$ | $32.8 \pm 24.3$ |

Table 14: Results from other work from similar user studies to ours, which is assessing the perceived ambiguity of prototypical parts. Results for ProtoPNet and ProtoConcepts are referenced from Ma et al. (2023), while findings for PIPNet and LucidPPN are cited from Pach et al. (2024), as indicated by the asterisk symbol (*).

| Method | Dataset | User Acc. [%] | p-value |
|---|---|---|---|
| ProtoPNet* | | $51.5 \pm 5.2$ | $0.288$ |
| ProtoConcepts* | CUB | $\mathbf{62.1 \pm 5.4}$ | $\mathbf{3 \cdot 10^{-5}}$ |
| PIP-Net* | | $\mathbf{60.0 \pm 18.1}$ | $\mathbf{0.002}$ |
| LucidPPN* | | $\mathbf{67.9 \pm 16.9}$ | $\mathbf{2 \cdot 10^{-6}}$ |

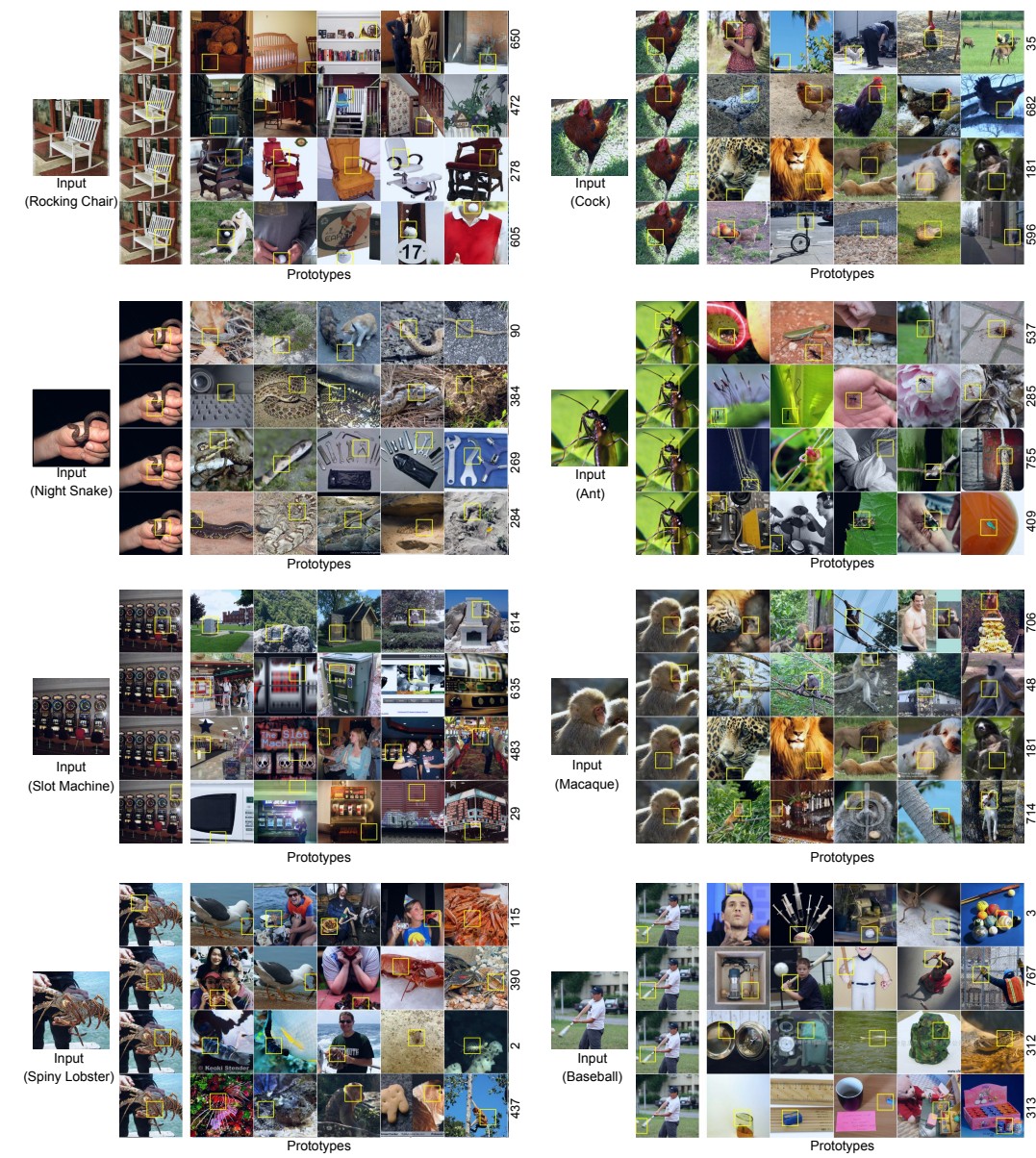

Figure 19: The figure demonstrates the prototypical explainability of InfoDisent(Swin-S) model by visualizing prototypes generated for arbitrary ImageNet images, highlighting relevant image regions and their corresponding channel-specific prototypes.

**Disentanglement metrics** First we focus on the prototype diversity, we check what is the overlap of patches used to represent a prototype between different prototypes. Additionally, we've quantified prototype distribution using three key metrics, derived from the five most significant prototype activations per image:

- Intra-Class Diversity (0-1): Calculated from the mean normalized occurrence frequency of prototypes within their respective classes, averaged globally. This measures within-class prototype consistency.

- Inter-Class Diversity (0-1): Assesses prototype sharing between different classes by measuring their cross-class appearance frequency.

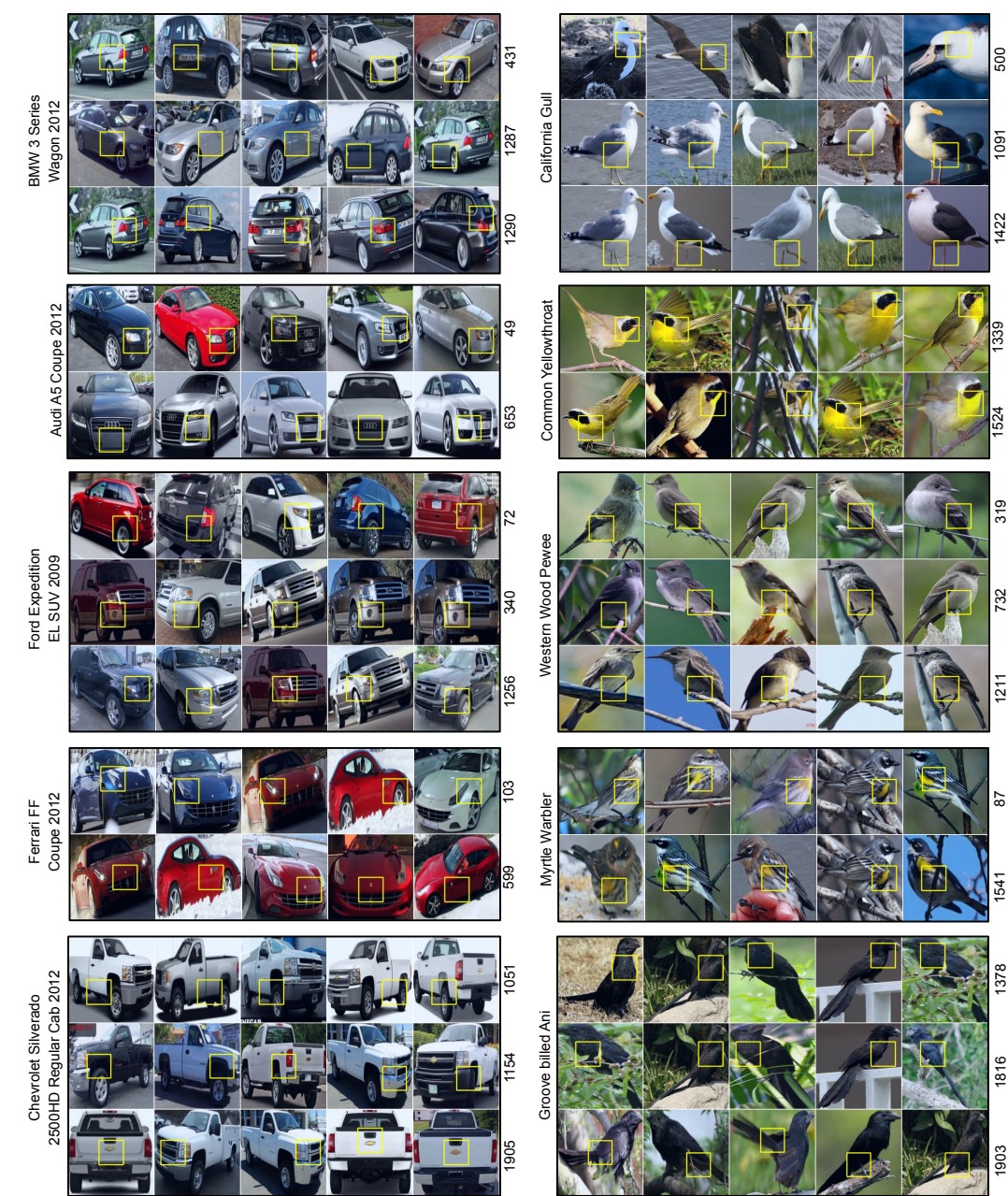

Figure 20: Prototypes of sample classes from InfoDisent(ResNet-50) model trained on the cropped Stanford Cars dataset (left) and the CUB-200-2011 dataset (right). Yellow frames highlight the class prototypes.

- Diversity Ratio: Computed as the ratio of Inter-Class to Intra-Class diversity. This normalized measure highlights the balance between consistent within-class prototypes and their specificity across classes.

Subsequently, we investigated the sparsity of activations and found that InfoDisent activates significantly fewer channels than its black-box baseline.

Theoretically, this sparsity stems from InfoDisent's frozen backbone and disentangling matrix, which effectively prune non-discriminative pathways and encourage concept localization. This syn-

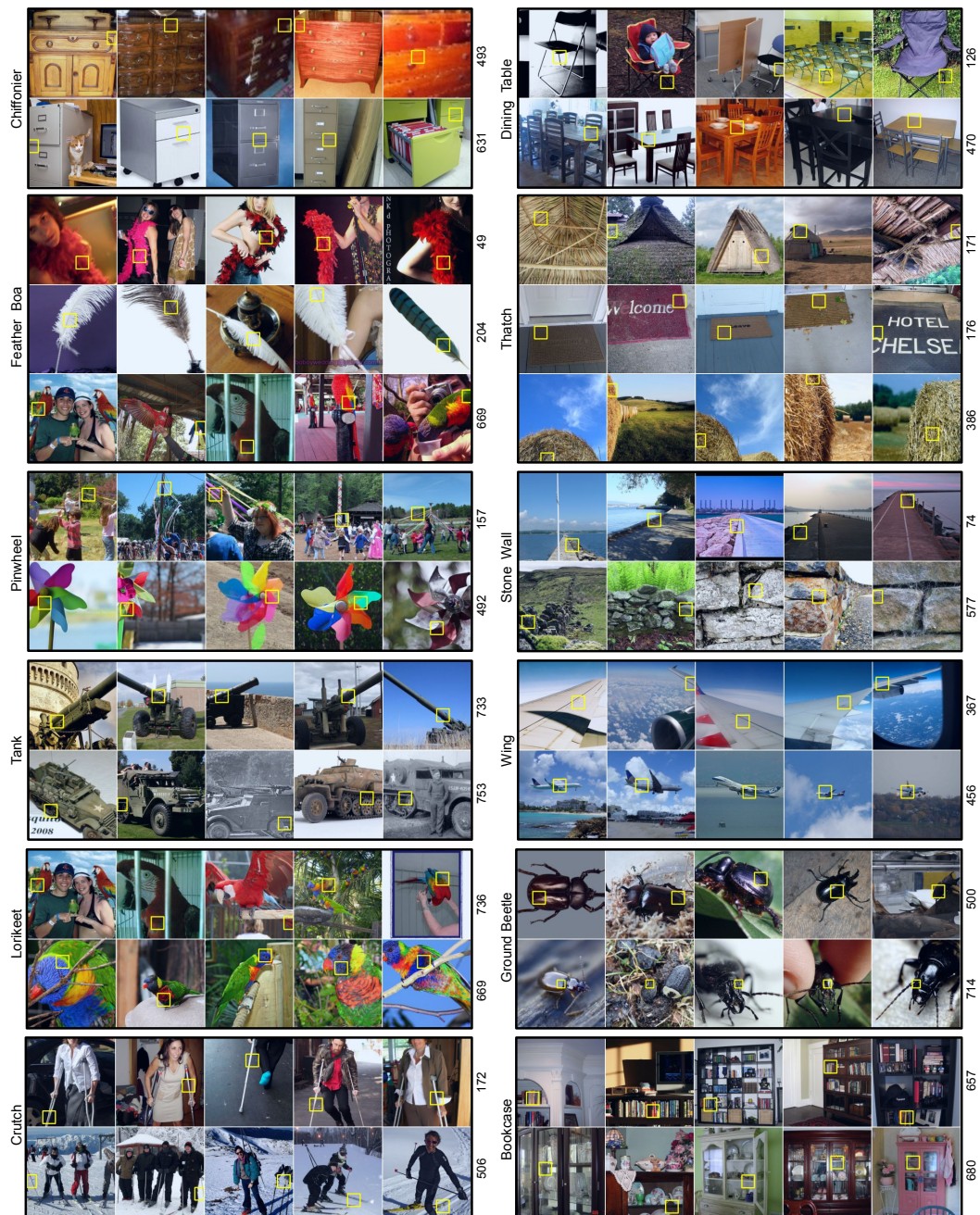

Figure 21: Visualizations of InfoDisent(ViT-B/16) model-generated prototypes for selected ImageNet categories.

ergy between sparsity and disentanglement not only enhances interpretability by simplifying concept attribution but also maintains scalability.

## D   MEDICAL DATASET EXPERIMENT

To validate the versatility of InfoDisent, we applied it to medical dataset of chest X-Rays to identify patients with pneumonia Kermany (2018). Below we present both, results of numerical experiments (accuracy) in Tab. 17 and exemplary explanations Fig. 27a for healthy patients and Fig. 27b for sick patients. One can observe, that prototypical explanations focus on different anatomical locations of

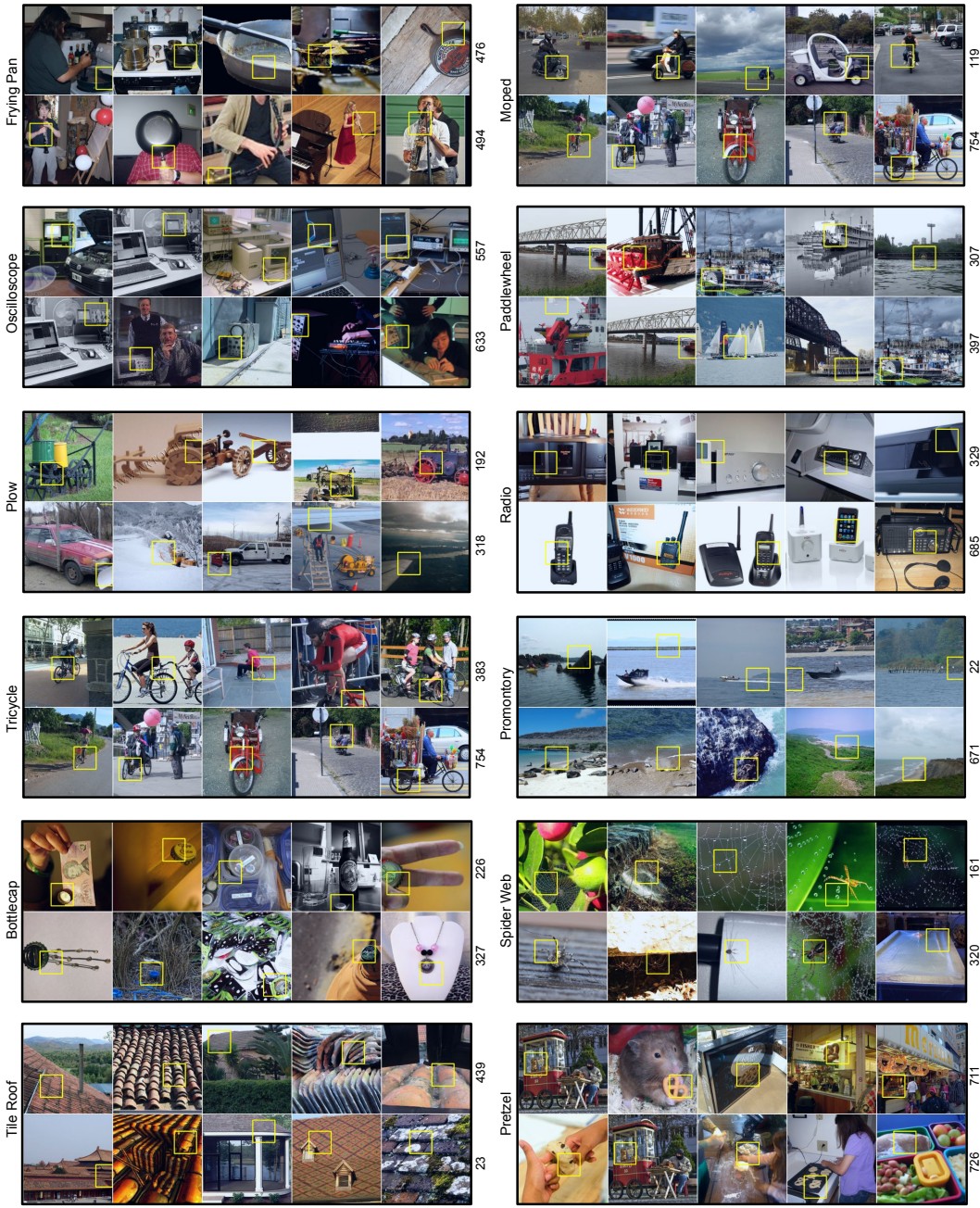

Figure 22: Representative ImageNet class prototypes produced by the InfoDisent(Swin-S) model.

the lungs, while performance of the model is slightly worse than its black-box counterpart (ResNet-18). Moreover, InfoDisent operates on ImageNet-pretrained backbone, while baseline ResNet-18 is unfrozen during training.

# E    LLM USAGE

We have utilized LLMs such as ChatGPT and Gemini to refine the grammar, identify typos and polish the writing style of this work.

Table 9: Detailed results on FunnyBirds.

| Backbone | Method | ACC | B.I. | Com. | Cor. | Con. | mX |
|---|---|---|---|---|---|---|---|
| ViT-B/16 | IG | 0.98 | 1.00 | 0.88 | 0.65 | 0.91 | 0.82 |
| | IG abs. | 0.98 | 1.00 | 0.92 | 0.63 | 0.74 | 0.76 |
| | RISE | 0.98 | 1.00 | 0.78 | 0.79 | 0.75 | 0.77 |
| | LIME | 0.98 | 1.00 | 0.90 | 0.00 | 0.00 | 0.30 |
| | IxG | 0.98 | 1.00 | 0.54 | 0.51 | 0.67 | 0.57 |
| | Grad-CAM | 0.98 | 1.00 | 0.81 | 0.70 | 0.48 | 0.66 |
| | Rollout | 0.98 | 1.00 | 0.81 | 0.76 | 0.00 | 0.52 |
| | Chefer-LRP | 0.98 | 1.00 | 0.90 | 0.74 | 0.95 | 0.86 |
| | InfoDisent | 0.98 | 0.81 | 0.82 | 0.63 | 0.91 | 0.78 |
| ResNet50 | IG | 1.00 | 1.00 | 0.86 | 0.59 | 0.98 | 0.81 |
| | IG abs. | 1.00 | 1.00 | 0.87 | 0.53 | 0.86 | 0.75 |
| | RISE | 1.00 | 1.00 | 0.70 | 0.56 | 0.61 | 0.62 |
| | LIME | 1.00 | 1.00 | 0.86 | 0.00 | 0.00 | 0.29 |
| | IxG | 1.00 | 1.00 | 0.58 | 0.54 | 0.80 | 0.64 |
| | Grad-CAM | 1.00 | 1.00 | 0.74 | 0.55 | 0.78 | 0.69 |
| | B-cos | 0.96 | 0.87 | 0.89 | 0.69 | 0.89 | 0.82 |
| | X-DNN | 0.99 | 1.00 | 0.91 | 0.60 | 0.87 | 0.79 |
| | InfoDisent | 1.00 | 1.00 | 0.91 | 0.65 | 0.97 | 0.84 |

Table 10: Performance under Image Perturbations. Note that J stands for Jaccard Similarity Index, and C stands for Cosine Similarity.

| Data | Model | Param. | Brightness | | Contrast | | Gaussian | | Saturation | |
|---|---|---|---|---|---|---|---|---|---|---|
| | | | (J) | (C) | (J) | (C) | (J) | (C) | (J) | (C) |
| CARS | ConvNeXt-Tiny | 0.01 | 0.9921 | | 0.9997 | | 0.9903 | | 0.9996 | |
| | | 0.1 | 0.9552 | | 0.9985 | | 0.9628 | | 0.9987 | |
| | | 0.5 | 0.8640 | | 0.9921 | | 0.8830 | | 0.9933 | |
| | DenseNet121 | 0.01 | 0.9979 | | 0.9999 | | 0.9981 | | 0.9999 | |
| | | 0.1 | 0.9859 | | 0.9997 | | 0.9756 | | 0.9997 | |
| | | 0.5 | 0.9273 | | 0.9985 | | 0.9271 | | 0.9985 | |
| | ResNet50 | 0.01 | 0.9961 | | 1.0000 | | 0.9948 | | 1.0000 | |
| | | 0.1 | 0.9825 | | 0.9998 | | 0.9835 | | 0.9998 | |
| | | 0.5 | 0.9291 | | 0.9993 | | 0.9314 | | 0.9990 | |
| CUB | ConvNeXt-Tiny | 0.01 | 0.9890 | | 0.9992 | | 0.9824 | | 0.9987 | |
| | | 0.1 | 0.9531 | | 0.9967 | | 0.9607 | | 0.9971 | |
| | | 0.5 | 0.8463 | | 0.9856 | | 0.8521 | | 0.9865 | |
| | DenseNet121 | 0.01 | 0.9969 | | 0.9995 | | 0.9912 | | 0.9994 | |
| | | 0.1 | 0.9735 | | 0.9992 | | 0.9735 | | 0.9990 | |
| | | 0.5 | 0.9101 | | 0.9964 | | 0.9022 | | 0.9960 | |
| | ResNet50 | 0.01 | 0.9888 | | 0.9998 | | 0.9880 | | 0.9998 | |
| | | 0.1 | 0.9666 | | 0.9992 | | 0.9673 | | 0.9993 | |
| | | 0.5 | 0.8983 | | 0.9966 | | 0.8963 | | 0.9966 | |

Table 11: Distribution of answers in user study on user confidence in model's prediction based on explanation.

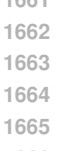

| Dataset | Fairly ... correct | Somewhat ... correct | Somewhat ... incorrect | Fairly ... incorrect |
|---|---|---|---|---|
| CUB | 585 | 316 | 208 | 91 |
| ImageNet | 449 | 248 | 189 | 314 |

Table 12: Distribution of answers in user study on prototypical part disambiguity.

| Dataset | Correct | Incorrect |
|---------|---------|-----------|
| CUB | 776 | 424 |
| ImageNet | 712 | 488 |

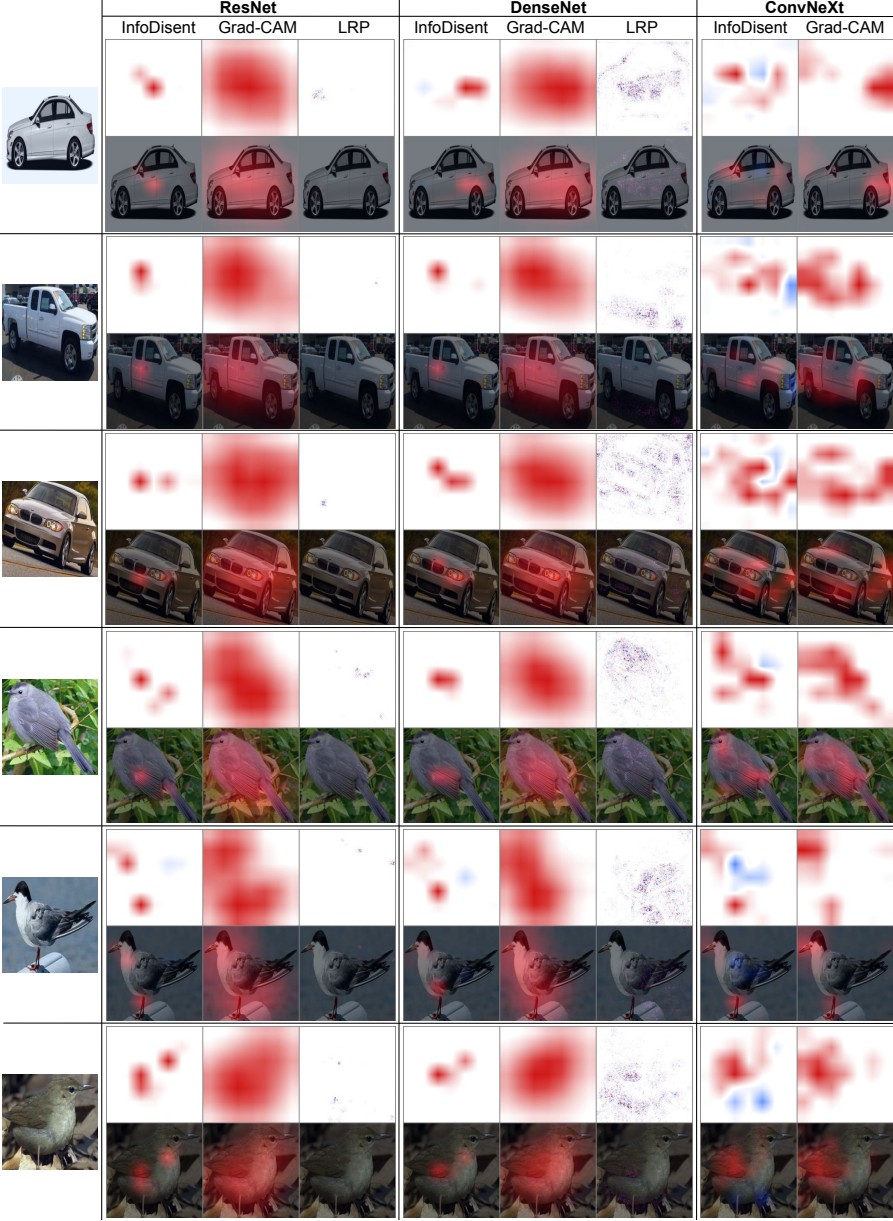

Figure 23: Comparison of example heat maps generated by InfoDisent proposed model and competing approaches. Our technique produces more focused heat maps by leveraging representational channels and the information bottleneck principle, outperforming traditional methods like Grad-CAM. InfoDisent maps channel activations from the last but one layer. These activations highlight precise and important regions of the input. Their interpretation aligns directly with other attribution-based approaches we compare against, such as LRP and Grad-CAM.

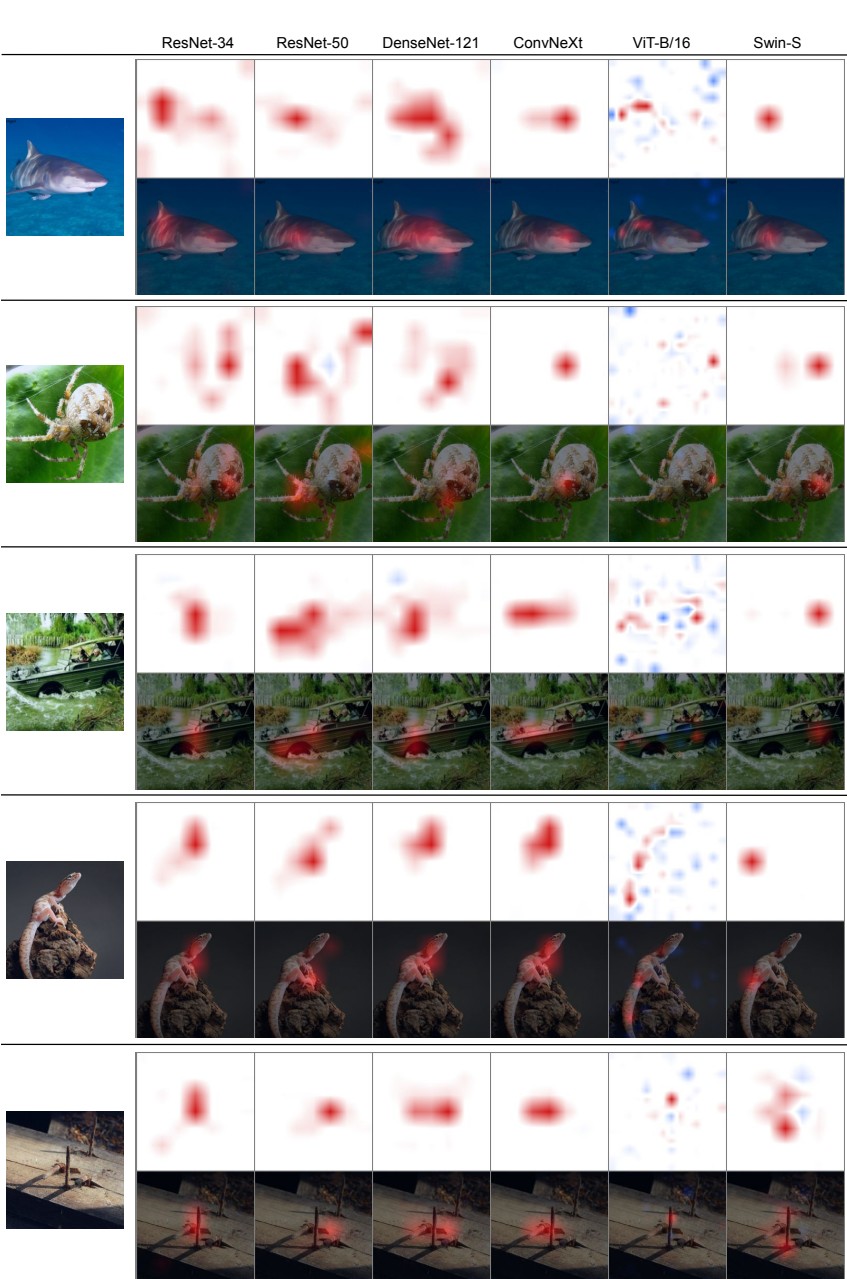

Figure 24: Example heatmaps generated by InfoDisent, our proposed method, demonstrating activation regions on sample photos from the ImageNet test set.

An image has been classified by the model. Below the image there is an explanation that the model gave to justify its decision. Based on the explanation, what do you think about the model's prediction?

A. Fairly confident that the model is correct.
B. Somewhat confident that the model is correct.
C. Somewhat confident that the model is incorrect.
D. Fairly confident that the model is incorrect.

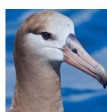

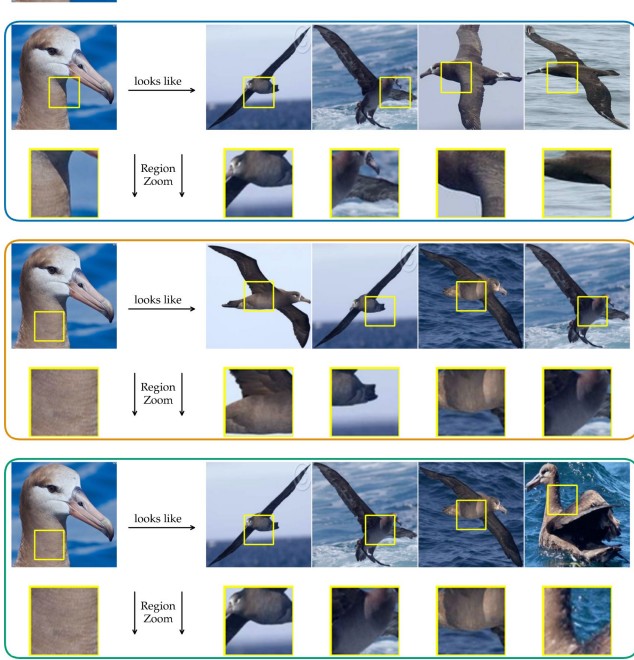

Figure 25: An exemplary question from the user study on user confidence.

Table 15: Disentanglement metrics.

| Dataset | Model | Intra-Class | Inter-Class | Diversity |
|---|---|---|---|---|
| CAR | ConvNeXt-Tiny (InfoD) | 0.261 | 0.033 | 0.132 |
| | ConvNeXt-Tiny (Base) | 0.218 | 0.040 | 0.188 |
| | DenseNet121 (InfoD) | 0.639 | 0.011 | 0.018 |
| | DenseNet121 (Base) | 0.371 | 0.014 | 0.039 |
| | ResNet50 (InfoD) | 0.741 | 0.010 | 0.014 |
| | ResNet50 (Base) | 0.428 | 0.013 | 0.031 |
| CUB | ConvNeXt-Tiny (InfoD) | 0.253 | 0.021 | 0.087 |
| | ConvNeXt-Tiny (Base) | 0.169 | 0.016 | 0.098 |
| | DenseNet121 (InfoD) | 0.562 | 0.011 | 0.021 |
| | DenseNet121 (Base) | 0.337 | 0.015 | 0.047 |
| | ResNet50 (InfoD) | 0.617 | 0.017 | 0.029 |
| | ResNet50 (Base) | 0.369 | 0.015 | 0.041 |
| ImageNet | ConvNeXt-L-InfoD | 0.285 | 0.022 | 0.081 |
| | ConvNeXt-L (Base) | 0.149 | 0.018 | 0.124 |
| | DenseNet121 (InfoD) | 0.334 | 0.012 | 0.038 |
| | DenseNet121 (Base) | 0.277 | 0.012 | 0.046 |
| | Swin-S (InfoD) | 0.393 | 0.011 | 0.028 |
| | Swin-S (Base) | 0.159 | 0.011 | 0.074 |

Figure 26: An exemplary question from the user study on disambiguity of prototypical parts.

Table 16: Number of channel used to make predictions. One can observe that InfoDisent is much sparser than a baseline model.

| Dataset | Model | Baseline | InfoDisent |
|---------|-------|----------|------------|
| CARS | ConvNeXt-Tiny | 380 ± 8.6 | 246 ± 8.1 |
| | DenseNet121 | 214.1 ± 19.9 | 28.1 ± 12.1 |
| | ResNet50 | 868.2 ± 77.3 | 111 ± 113.2 |
| CUB | ConvNeXt-Tiny | 466.1 ± 71 | 99.2 ± 9.7 |
| | DenseNet121 | 208.7 ± 26.4 | 37.7 ± 30.1 |
| | ResNet50 | 961.9 ± 99.3 | 187.3 ± 249.6 |
| ImageNet | ConvNeXt-L | 764.2 ± 12.4 | 495.7 ± 35.7 |
| | DenseNet121 | 254.3 ± 17.1 | 112.8 ± 103.6 |
| | MaxVit | 261.4 ± 12.3 | 250.2 ± 10.6 |
| | ResNet50 | 1049 ± 59.3 | 226.7 ± 240.6 |
| | Swin-S | 455.9 ± 11 | 328.3 ± 26.9 |
| | ViT-B/16 | 16472.4 ± 6.7 | 294 ± 34.3 |

| Model | Accuracy [%] |
|-------|--------------|
| InfoDisent | 90.55 ± 0.45 |
| ResNet-18 | 92.57 ± 1.14 |

Table 17: Pneumonia detection performance comparison. We compare our proposed InfoDisent method (using a frozen, ImageNet-pretrained backbone) against a standard black-box ResNet-18 (fully unfrozen). InfoDisent achieves results comparable to, though slightly lower than, the fully fine-tuned baseline.

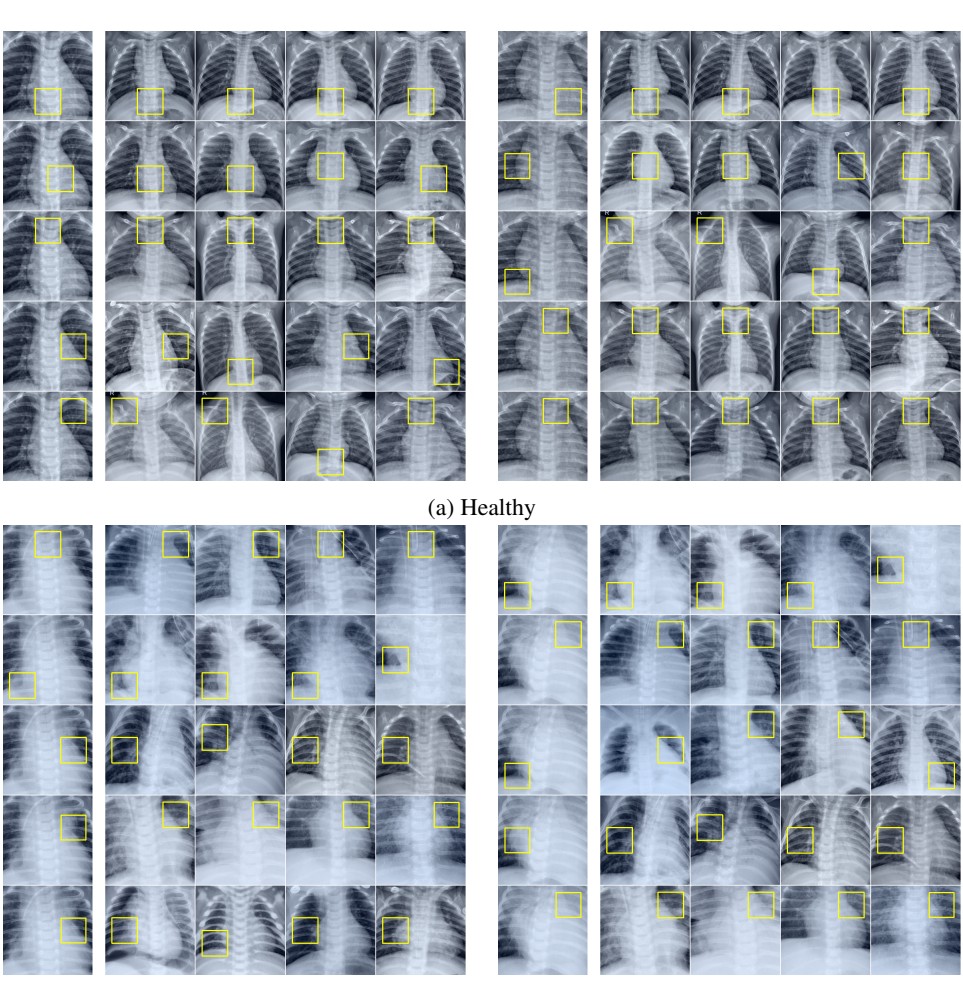

(a) Healthy

(b) Pneumonia

Figure 27: Exemplary explanation for pneumonia detection based on  Kermany (2018) dataset.

