# OpenReview forum: "InfoDisent: Explainability of Image Classification Models by Information Disentanglement"
_ICLR.cc/2026/Conference — Submitted to ICLR 2026_

### Official Review · Reviewer_WmEX · 2025-10-30

**Soundness:** 3
**Presentation:** 1
**Contribution:** 2
**Rating:** 4
**Confidence:** 4

**Summary:**

The paper presents InfoDisent, a post-hoc explainability method that disentangles features from a frozen pretrained backbone through an orthogonal decomposition of the channels. The goal is to identify prototypical parts that explain the model's decisions. The paper provides multiple experiments showcasing that the approach is applicable to a wide range of models, from CNNs to Transformer architectures, with classification performance on several datasets. It also includes an evaluation of interpretability on the FunnyBirds dataset and two user studies assessing confidence in predictions and ambiguity in explanations.

**Strengths:**

- **Significance**:
	- The proposed approach is flexible and can be effectively applied to various pretrained backbones, which is a significant advantage over methods limited to specific architectures or that needs to retrain the model.
	- The results demonstrate that the method largely preserves the accuracy of the original pretrained backbone, especially compared to some other prototypical part methods.
- **Originality**: The decomposition into prototypical parts through the orthogonal matrix and its optimization process, is original. The reformulation of the optimization problem using skew-symmetric matrices is clever and an interesting technical contribution, although hidden  and only mentioned in the appendix.

**Weaknesses:**

- **Clarity**:
	- The paper's overall structure and clarity needs to be improved. Several key discussions and technical details, such as the full optimization process, are briefly mentioned or only detailed in the appendix. Describing the overall training and optimization process more comprehensively in the main paper would significantly enhance understanding.
	- The positioning of tables and figures within the paper is confusing, and detracts from the reading flow.
- **Quality**: The supplementary material includes a section labeled "ablation study" which, in its current form, does not present a true ablation of the method's components. The method involves multiple design choices beyond a "default classification head", the orthogonal decomposition, max pooling, and the non-negative coefficients matrix. A proper ablation study for each of these components is missing, making it difficult to assess their individual importance and contribution to the method.
- **Significance**: While the method offers flexibility across architectures, the quality of interpretation, from the evidence presented, appears to be similar to existing prototypical part methods. The paper could better articulate how InfoDisent's interpretability provides a distinct advantage or deeper insights beyond its architectural flexibility.

**Questions:**

- How consistent are the prototypes found across different classes and samples? Does the same channel consistently link to the same "concept" or "prototype," or do these interpretations vary significantly?
- It is mentioned in the section about "optimizing U and W" that the goal is to find a product of a sparse and unitary matrix. Are there any explicit sparsity constraints or regularization terms added to the optimization objective, other than optimizing for an orthogonal matrix, to encourage this desired sparsity?
- Can the method be effectively applied when transferring from one dataset to another (e.g., using a backbone trained on ImageNet and applying InfoDisent to explain predictions on a medical imaging dataset)?

---

> ### Author Response · Authors · 2025-11-21
>
> > How consistent are the prototypes found across different classes and samples? Does the same channel consistently link to the same "concept" or "prototype," or do these interpretations vary significantly?
>
> When it comes to the consistency of prototypes between classes and samples, in Table 15 in the Appendix we provided metrics related to diversity of concepts. We can observe that diversity is low within the same data class, as well as between data classes, which shows that in most cases the same channel links to the same concepts, as they are very similar.
>
> > It is mentioned in the section about "optimizing U and W" that the goal is to find a product of a sparse and unitary matrix. Are there any explicit sparsity constraints or regularization terms added to the optimization objective, other than optimizing for an orthogonal matrix, to encourage this desired sparsity?
>
> There are no explicit sparsity constraints or additional regularization terms applied to the optimization objective beyond optimizing for orthogonal transformations.
>
> The sparsity in information flow through channels is enforced solely through the network architecture itself, specifically via the "ArgMax Over Channels" operation shown in Figure 2. The orthogonal transformation (unitary matrix) is optimized to maintain important properties like preserving norms and preventing information collapse, but the actual sparsity emerges as a consequence of the architectural design rather than being explicitly constrained in the optimization objective. This approach separates the concerns: orthogonal transformations handle the mathematical properties of the transformations, while the network architecture handles the sparsity through the ArgMax operation.
>
> > Can the method be effectively applied when transferring from one dataset to another (e.g., using a backbone trained on ImageNet and applying InfoDisent to explain predictions on a medical imaging dataset)?
>
> Additionally, we will perform experiments on medical dataset, which we report during the discussion period as it is computationally expensive to perform in such a short timeframe. We plan to apply InfoDisent to  https://huggingface.co/datasets/hf-vision/chest-xray-pneumonia

---

> > ### Author Response · Authors · 2025-11-28
> > **Medical Dataset Experiment Ready**
> >
> > We appreciate the suggestion to evaluate our approach on a different domain. In the revised manuscript, we have included results on the [Kermany (2018)] medical imaging dataset.
> >
> > InfoDisent, utilizing a frozen ResNet-18 backbone pretrained on ImageNet, achieved 90% accuracy. This is highly competitive with a fully fine-tuned ResNet-18 (unfrozen backbone), which achieved 92% accuracy.
> >
> > The fact that InfoDisent performs comparably to the fine-tuned baseline, without requiring backbone weight updates, demonstrates the robustness of the learned representations and highlights the method's versatility for transfer learning tasks in specialized domains like medical imaging.

---

### Official Review · Reviewer_xG6k · 2025-10-31

**Soundness:** 2
**Presentation:** 2
**Contribution:** 2
**Rating:** 2
**Confidence:** 4

**Summary:**

The proposed method InfoDisent is a post-training classification head designed to enhance the interpretability of frozen image-based model backbones.
The underlying idea that pretrained features already contain the necessary atomic concepts, is plausible and well motivated.
The method combines an orthogonal transformation, sparse pooling of the most positive and negative activations per channel, and a nonnegative linear classifier, to isolate channel-wise representations that correspond to salient semantic cues.
The authors interpret this mechanism as an information bottleneck that disentangles latent features into concept-like channels and enables prototype-based explanations in a single forward pass.
Experiments on several vision datasets demonstrate comparable accuracy and visually coherent explanations.

**Strengths:**

The proposed method is technically simple and broadly applicable. It can be attached to any frozen backbone without finetuning, offering post-training interpretability with minimal computational overhead. The model produces explanations in a single forward pass, which is computationally efficient compared to gradient-based saliency approaches and aligns with the idea of self-explainable architectures. The qualitative results are intuitive and visually consistent across datasets, and the experiments demonstrate reasonable generality of the method.

**Weaknesses:**

**1.**
L57: The paper claims that InfoDisent “leverages the information bottleneck principle” to ensure each channel encodes an atomic concept, but no mutual information quantity (e.g., (I(X;T)), (I(T;Y))) is defined or measured. The idea of “information bottleneck” is used rhetorically, not mathematically.

**2.**
L155: The method section introduces “extremely sparse pooling” and “unitary map (U)” yet provides no justification for how these operations realize an information bottleneck. There is also no ablation showing the effect of each component.

**3.**
L164: The explanation of sparse pooling uses notation (K \to \text{mx_pool}(K) = \max(\text{ReLU}(K)) - \max(\text{ReLU}(-K))). The presentation is algebraically correct but conceptually vague.

**4.**
L175: The notation (I = (I_{rs})*{rs} \to J = (U I*{rs}) \to v_J = \text{mx_pool}(J)) is confusing. Indices (I, J, r, s) are inconsistently used to denote both spatial positions and transformed tensors. This ambiguity makes even simple tensor operations difficult to follow.

**5.**
L64, Table 3: ImageNet experiments are introduced as the first generalization of prototypical parts to large-scale datasets, yet Table 3
shows a performance drop. Although the performance–explainability tradeoff is understandable, the observed performance gap between CNN and ViT backbones on ImageNet should be further investigated. The method yields a noticeably larger accuracy drop for CNNs, while transformers retain most of their baseline performance. This difference likely reflects how the proposed orthogonal transformation and sparse pooling interact with backbone representations, but the paper provides no analysis or hypothesis. A deeper examination of this effect would strengthen the claims about scalability and general applicability.

**6.**
Much of the essential methodological and quantitative content resides in the appendix, while the main text remains largely descriptive. Core analyses, e.g.,  disentanglement metrics and quantitative evaluation of interpretability, are only referenced but not summarized. As a result, the paper’s main body does not provide enough self-contained evidence for the claims. The authors should bring key results and analyses into the main text to ensure scientific transparency.

**7.**
Heatmaps are compared qualitatively, but the text asserts that “our approach yields more focused heatmaps.” This claim is unsupported by any numeric localization score or overlap measure.

**8.**
L1509: Theoretical explanation of sparsity seems important but no ablation or analysis demonstrates causal relation between orthogonalization and sparsity. Orthogonalization alone does not guarantee sparsity, and the claim remains speculative without quantitative evidence.

**9.**
The claimed novelty of InfoDisent is somewhat overstated. Similar prototype-based interpretability approaches already explore post-hoc or semi-post-hoc explanations using frozen backbones. The main contribution here lies in a streamlined architectural design, which is technically neat but conceptually incremental. The introduction would benefit from a clearer articulation of what gap in interpretability research this specific formulation addresses.

**10.**
The term “atomic concept” is commonly used in the interpretability literature to denote minimal, human-recognizable visual units. I accept this usage as conventional. However, in this paper, the notion is treated as an achieved property rather than a descriptive goal. The authors should clarify that “atomic” here refers to qualitative observations, not a formally verified characteristic, and avoid implying theoretical guarantees.

**11.**
Figures are numerous but lack quantitative caption summaries. The prose often mixes conceptual justification with qualitative description, making it difficult to distinguish evidence from interpretation.

**Questions:**

**Q1.**  You state that InfoDisent “leverages the information bottleneck principle,” yet no mutual information quantity (e.g., (I(X;T)), (I(T;Y))) is defined or measured.
Can you formally specify what information quantity is being constrained, and how your loss function or architecture approximates an information bottleneck objective?
If the term is used metaphorically, please clarify this explicitly.


**Q2.**
The method section introduces the “extremely sparse pooling” and “unitary map (U)” but provides no justification for how these operations achieve an information bottleneck.
Could you provide a theoretical argument or an ablation experiment showing how removing either component affects information compression or interpretability?


**Q3.**
Your sparse pooling function (L165) selects only two scalars per channel.
What statistical or information-theoretic reasoning supports this choice?
Have you compared it to alternatives such as top-(k) pooling or soft log-sum-exp pooling, and can you report how these affect sparsity and accuracy?


**Q4.**
The notation in L177-184 is hard to follow.
Can you provide a clearer tensor formulation, explicitly defining the dimensions of each variable to make the method reproducible?


**Q5.**
The ImageNet experiments show a noticeable performance drop, particularly for CNNs.
Although some tradeoff between interpretability and accuracy is expected, could you investigate why the degradation differs between CNN and ViT backbones?
For example, does orthogonal transformation interact differently with convolutional versus attention features?
Please include a diagnostic or ablation study to analyze this effect.


**Q6.**
You claim that InfoDisent “matches classical models,” but no error bars or variance estimates are shown.
Could you report statistical confidence intervals or repeat runs to substantiate the claim of comparable performance?


**Q7.**
Much of the methodological and quantitative content is relegated to the appendix.
Can you summarize key results (e.g., disentanglement metrics, quantitative interpretability scores) in the main paper to make it self-contained?
In particular, which quantitative evidence best supports your central claims?


**Q8.**
The text claims that InfoDisent produces more focused heatmaps than Grad-CAM or LRP, but no numeric localization or overlap metrics are shown.
Could you provide quantitative comparisons (e.g., IoU, pointing-game accuracy, or deletion/insertion scores) to support this statement?


**Q9.**
You attribute sparsity to the combination of a frozen backbone and an orthogonal transformation, but no analysis demonstrates this causality.
Could you add an ablation experiment showing activation sparsity with and without orthogonalization, or visualize activation histograms to quantify the effect?


**Q10.**
Prototype-based interpretability methods such as ProtoPNet and PIP-Net already explore post-hoc interpretability using frozen backbones.
Can you clarify what precise gap InfoDisent fills beyond simplified architecture?
In what aspect does your approach advance the understanding of interpretability rather than streamline prior designs?


**Q11.**
The paper frequently refers to “atomic concepts” as if they are achieved properties.
Could you clarify how you define atomicity in measurable terms?
Is there any quantitative analysis (e.g., concept purity, mutual information, or ablation sensitivity) showing that channels correspond to minimal, non-overlapping semantic units?

---

> ### Author Response · Authors · 2025-11-21
>
> > Q1. Information Bottleneck
>
> The term Information Bottleneck (IB) is used to describe an architectural constraint implemented through sparsity and orthogonality in the proposed head, as detailed in Section 3. The objective of InfoDisent is to compress information within the bottleneck layer, but it does not involve the direct optimization of the specific mutual information objective $I(X;T) - \beta I(T;Y)$.
>
> > Q2. Justification for Pooling and Unitary Map
>
> The information bottleneck effect is achieved through the combination of the extremely sparse pooling and the unitary map (U). An Ablation Study presented in the Supplement demonstrates that the InfoDisent channels exhibit significantly sparser usage compared to standard models, supporting their role in information compression.
>
> > Q3. Rationale for Pooling
>
> The choice to select only two scalars per channel in the sparse pooling function was made because these extreme values effectively capture the distribution of feature values. If there is minimal difference between the two extreme values, the concept activation will be low; otherwise, it will be high, which follows the intuition of Focal Similarity from ProtoPool. Alternative methods like top-$k$ or soft log-sum-exp pooling were not compared because they would unnecessarily increase the architectural complexity, potentially hindering interpretability by making the network's reasoning more complex.
>
> > Q4. Tensor Formulation
>
> The full method is made reproducible through the provision of code. Explicitly adding dimensions to the equations in the main text was avoided as it was deemed likely to make the notation more confusing and complex rather than clarifying it.
>
> > Q5. Performance Degradation in CNN vs. ViT
>
> The varying impact of orthogonalization on CNNs and ViTs has been previously studied, and conclusions from those studies were integrated into the main body of the work. It is hypothesized that the performance gap (e.g., a larger gap for ViTs in fine-grained classification) depends on the task and the suitability of the architecture for that task. A large-scale investigation into the general properties of CNNs and ViTs under orthogonalization is considered outside the scope of the current work.
>
> > Q6. Statistics for Performance Claims
>
> The reported results are the average of over 3 seeds. Deviations will be provided soon.
>
> > Q7. Key Quantitative Evidence
>
> The most important findings from the Appendix have been summarized in the new version of the main work. Quantitative evidence supporting the central claims includes:
> High Compactness metric (e.g., in FunnyBirds experiments), which supports the claim of more focused heatmaps.
> Sparsity of InfoDisent (referenced by Table 16 and Ablation Studies), which supports the claim of channel sparsity.
> Results from Spatial Misalignment experiments, which confirm that concepts do not rely heavily on the full object, supporting the part-based interpretation.
> Diversity comparisons, which show that the learned concepts are coherent.
>
> > Q8. Quantitative Support for Heatmap Focus
>
> Quantitative justification for InfoDisent producing more focused heatmaps is provided by the Compactness metric results, particularly in the FunnyBirds experiments. InfoDisent achieves a higher Compactness metric than methods like Grad-CAM, indicating superior localization and focus.
> > Q9. Causality of Sparsity
>
> An ablation experiment specifically demonstrating the causality of sparsity with and without orthogonalization was performed and is presented in the Ablation Study Section in the Appendix as well as in Table 16.
>
> > Q10. Gap Filled by InfoDisent
>
> Existing prototypical part-based interpretability methods fall into two categories: training-time methods (e.g., ProtoPNet, PIP-Net) that modify and unfreeze the backbone, and post-hoc methods (e.g., Grad-CAM) that do not train any part of the network.
> InfoDisent fills a methodological gap by offering a hybrid approach. It leaves the backbone frozen but introduces a trainable, interpretable head that transforms (disentangles) the features to create a prototypical-parts-based representation. This advancement allows for gaining a prototypical parts representation by only training an efficient head on top of a frozen backbone, thus eliminating the need to train or alter the full model.
>
> > Q11. Defining and Measuring Atomicity
>
> Atomic concepts are defined as concepts that share similar semantics and represent parts of an object rather than the entire object. Quantitatively, this property is supported by:
> Results from the Spatial Misalignment analysis, confirming that the concepts are not heavily reliant on the full object.
> Diversity comparison results, showing concept coherence.
> The model's design, which allows concepts to correspond to a single patch from a feature map, preventing them from covering a full object.
> Quantitative evidence of sparsity is reported in Table 16.

---

> > ### Comment · Reviewer_xG6k · 2025-11-27
> >
> > Thank you for your answers.
> >
> > I will carefully consider your response in reaching the final decision. Please incorporate these new findings into the manuscript.
> >
> > Best regards,
> > Reviewer xG6k

---

> > > ### Author Response · Authors · 2025-11-28
> > >
> > > Dear Reviewer,
> > >
> > > We have added deviations to the revised version of the work in Tables 1, 2 and 3.
> > >
> > > Best Regards,
> > > The Authors

---

### Official Review · Reviewer_FCXn · 2025-10-31

**Soundness:** 2
**Presentation:** 2
**Contribution:** 2
**Rating:** 2
**Confidence:** 4

**Summary:**

The paper introduces InfoDisent, a method for explainability (XAI) that disentangles information in the final layer of pretrained models into interpretable atomic concepts. It’s a hybrid approach combining the flexibility of post-hoc methods with the concept-based transparency of ante-hoc models (like ProtoPNet). The method applies an orthogonal transformation and information bottleneck (using sparse pooling) to produce prototype-like channel activations without retraining the backbone. It demonstrates interpretability and competitive performance on datasets including CUB-200-2011, Stanford Cars, Stanford Dogs, and ImageNet, and includes user studies for human interpretability.

**Strengths:**

1. It unifies the interpretability of ProtoPNet-like models with the flexibility of post-hoc XAI, taking the best of the both worlds.
2. Evaluation spans five datasets, including the challenging ImageNet, where most prototype-based methods fail.

**Weaknesses:**

1. First of all, the authors should cite concept bottleneck models - related papers. That includes fully interpretable concept bottleneck models to post hoc-based concept bottleneck models.

2. One big critcism is how to relate these prototypes to human-readable concepts. Often, these prototypes resemble to something unusual which does not match any known concept. I am coming from a medical imaging background, and I have seen this a lot in medical images where these prototypes often select a patch that is not related to any human anatomy. The interpretability community need to solve this challenge to actually produce plausible interpretation of a deep model. This is both true at a local and global level.

3. The paper's central concept, "information disentanglement", is not well-defined or rigorously evaluated. The authors claim the channels represent "atomic concepts" , but this is a subjective claim based on cherry-picked qualitative examples (e.g., Fig 1's "strawberry texture" ). The primary quantitative metric for disentanglement (RV coefficient) is relegated to the appendix (Table 14)  and shows mixed results, with InfoDisent performing worse than the baseline in several cases (e.g., ConvNeXt-L on ImageNet). For a paper with "Disentanglement" in the title, this core concept needs a much stronger theoretical grounding and more convincing quantitative support.

4. The paper introduces a trainable orthogonal transformation ($U$) and a sparse pooling operation, claiming this leads to "atomic concepts". However, it lacks a dedicated loss term to enforce this. There is no equation equivalent to a variational bottleneck's KL-divergence (like in $\beta$-VAE) or a mutual information maximization term that would mathematically compel the channels to represent independent, non-overlapping concepts.

5. The paper never formally defines what "disentanglement" means in this context. It is used as a qualitative descriptor for the desired outcome (interpretable prototypes), but the process of how the transform and bottleneck actually achieve this separation of information is not rigorously explained or validated.

6. The main paper presents Figure 2 as the model architecture. This figure is incomplete and misleading, as it shows a simple max(ReLU(...)) operation, which is non-differentiable and cannot be trained as-is. As you correctly point out, Figure 9 (in the Appendix) reveals the actual training architecture. SO, Fig 9 has to be in the main section.

7. The authors are not fully transparent about the performance-interpretability trade-off. Table 3  clearly shows that InfoDisent's accuracy is lower than the original (non-interpretable) backbone. For example, ResNet-50 drops from 76.1% to 67.8%, and Swin-S drops from 83.4% to 81.4%. This is a critical trade-off. However, the authors claim their approach "not only matches the performance of these classical models but also offers enhanced explainability". This is verifiably false; it doesn't match the performance. This trade-off is the cost of interpretability and should be discussed as such, not to be deemed as a non-issue. Also, Table 1 and Table 2 shows info-dissect falls short to many competitive methods.

**Questions:**

1. Could you please provide a formal, technical definition of "disentanglement" as it applies to this work?
2. There appears to be a significant discrepancy between the method's presentation in the main paper and the appendix. Figure 2 shows a simple, non-differentiable \textt{mx_pool} operation , while Figure 9 in the appendix reveals the actual, more complex training-time architecture, which relies on a Gumbel-Softmax estimator. This is a critical, non-obvious detail. Can you confirm that the Gumbel-Softmax-based architecture in Figure 9 is, in fact, the reproducible method, and will you move this to the main paper in a final version?

---

> ### Author Response · Authors · 2025-11-21
>
> > Formality of Disentanglement
>
> Our model incorporates two core mechanisms. The first is feature orthogonalization, which, analogous to Singular Value Decomposition (SVD), aims to decorrelate the feature channels, thereby forcing them to capture statistically independent factors of variation. The second mechanism, Argmax Over Channels, directly enforces the formation of "atomic concepts" by preserving only a single dominant activation value (considering both positive and negative activations) across an entire channel. Consequently, by the very definition of the argmax operation, the result is an "atomic" – that is, a single, dominant – concept per channel.
>
> The primary objective of this work is model interpretability, and we employ "information disentanglement" as a tool within our method to achieve this goal. Consequently, the paper focuses its evaluation primarily on experiments concerning model prediction explainability. As suggested, we have moved the RV coefficient results from Table 14 into the main body of the paper to give them appropriate prominence. Furthermore, in Table 16 we quantitatively demonstrate how our method utilizes a distinct and often more focused set of channels compared to the baseline, providing concrete support for the learned "atomicity."
>
> > Gumbel-Softmax Descritpion
>
> We thank the reviewer for their careful reading and constructive feedback. To address the identified weaknesses, we have made several changes to the manuscript. Specifically, we have updated Figure 2 to Figure 9, moved the explanation of the Gumbel-Softmax procedure to Section 3, and provided a definition of disentanglement in the Section 3.
>
> > Disentanglement Metrics
>
> Regarding the disentanglement metrics, we showcase the full results in Table 14 and Table 15 in the Supplementary Materials. While our model, InfoDisent, surpasses the baseline in multiple cases, we acknowledge that it does not do so universally. However, the primary aim of this work is to achieve better interpretability through the use of the disentanglement idea, a goal we have confirmed through our FunnyBirds experiments and the user study results. We have included extensive ablations and additional experimentations to thoroughly validate the model and transparently illustrate both the benefits and limitations of our approach. We prioritize transparent and reliable science by reporting all results as they are, including those few with a weaker RV coefficient, rather than selectively removing them. Fundamentally, we aimed for a novel interpretability architecture that is computationally efficient, scalable, and easily adaptable, which we believe we achieved. We demonstrate that the usage of a disentanglement technique can lead to more interpretable models, and we anticipate that further research, specifically focused on maximizing disentanglement, can elevate this model's metrics.
>
> > Citing Concept Bottlenecks
>
> We also added Concept Bottleneck references to the Related Works section, including recent advancements in their unsupervised and semi-supervised derivations.
>
> > Limitation of prototypical parts
>
> We agree with the reviewer that prototypical parts-based models currently lack a direct link to human-readable concepts; our user studies implicitly confirm this limitation. We view this as a limitation inherent to all prototypical parts methods and thus consider fixing it to be outside the scope of this work. However, very recent works show that it is possible to apply Vision-Language Models to acquire this human-understandable meaning (see https://openreview.net/pdf/077ff8e8f9fb964ac13f990ec90a5f3746235050.pdf). We have added a discussion of this point and its accompanying reference to the Limitations section.

---

> > ### Comment · Reviewer_FCXn · 2025-11-23
> > **Post-rebuttal**
> >
> > I thank all the authors for the rebuttal. I am still concerened with the prototypical parts. I saw the paper that the authors referenced and first of all that paper is still under review for ICLR 2026. Also, i have issues with that paper, so wont digress to that one. I believe if we truly want interptrability, the interpretable results must be human-interpretable, else the application will be limited.
> >
> > Regarding disentanglement, i still dont understand how true disentanglement happens in the model. Fundamentally, is it supervised or unsupervised disentanglement? if unsupervised, then its difficlt to achieve [1]
> >
> > [1] On the Impossibility of Learning Disentangled Representations Without Inductive Biases. Locatello et al. ICML 2019

---

> > > ### Author Response · Authors · 2025-11-26
> > > **post-rebuttal answer**
> > >
> > > Dear Reviewer,
> > >
> > > We thank you for the continued engagement and for raising these fundamental questions.
> > >
> > > 1. Regarding Disentanglement.
> > >
> > > You correctly point out the "Impossibility of Learning Disentangled Representations Without Inductive Biases" (Locatello et al., ICML 2019). We fully agree with this theoretical foundation: unsupervised disentanglement is impossible without priors.
> > >
> > > However, InfoDisent is not unsupervised. It falls under the umbrella of weakly-supervised disentanglement.
> > >
> > > As established in subsequent literature, specifically [1], the impossibility result of 2019 does not apply when auxiliary information (such as class labels) is available. In our framework, the "inductive biases" required to solve the disentanglement problem are provided by:
> > >
> > > Supervision via Class Labels: The model is trained to minimize classification loss. This forces the latent features to align with factors of variation that are discriminative for the specific classes (e.g., specific beak shapes or car headlights).
> > >
> > > Architectural Inductive Biases: We enforce orthogonality (analogous to PCA/SVD) and sparsity (via the bottleneck). These constraints force the model to distribute information into distinct, non-overlapping channels.
> > >
> > > Therefore, we do not claim to solve the unsupervised disentanglement problem. Rather, we leverage the classification task itself as the weak supervision to achieve disentanglement, which is theoretically sound and consistent with the findings in [1].
> > >
> > > 2. Regarding Prototypical Parts and Human Interpretability.
> > >
> > > We understand your concern, coming from a medical imaging background, that prototypes should ideally map to named semantic concepts (e.g., "edema" or "tumor") rather than just visual patches.
> > >
> > > However, we respectfully distinguish between visual interpretability (what part of the image the model looks at) and semantic labeling (naming that part).
> > >
> > > The concept of "Prototypical Parts" (visual patches acting as prototypes) is a widely accepted paradigm in the ML community, validated by top-tier publications including ProtoPNet (NeurIPS), ProtoTree (CVPR), and LucidPPN (ICLR). These methods define interpretability as "reasoning by visual similarity," which is valuable even without textual labels (e.g., showing a doctor which past case looks like the current patient, even if the feature is unnamed). Your criticisms about prototypical parts is general in the context of the XAI field, and this group of method. However, our contribution is addressing other limitations of prototypical parts, e.g. lack of generalizability to ImageNet, than lack of their human-level interpretability.
> > >
> > > While translating visual patches of prototypical parts into human language is an exciting direction (as discussed in the logic of the referenced ICLR 2026 submission / WACV work), requiring a prototypical network to inherently produce text-aligned concepts changes the scope of the work (to better human understanding of prototypical parts).
> > >
> > > References [1] Locatello, Francesco, et al. "Weakly-supervised disentanglement without compromises." International Conference on Machine Learning (ICML). PMLR, 2020.
> > >
> > > Best regards, The Authors

---

### Official Review · Reviewer_3ACv · 2025-11-02

**Soundness:** 3
**Presentation:** 3
**Contribution:** 3
**Rating:** 6
**Confidence:** 3

**Summary:**

The authors proposed InfoDisent, an interpretable image classification framework based on information disentanglement and sparse concept reasoning.
It applies an orthogonal transformation to decorrelate feature channels and a deterministic max–min pooling mechanism to extract positive and negative evidential activations, forming disentangled atomic concepts.
These concepts serve as interpretable units that link model decisions to localized object parts while maintaining scalability through channel sparsity and a frozen backbone.
Unlike conventional prototype models that rely on large prototype sets or end-to-end fine-tuning, InfoDisent achieves interpretability and efficiency simultaneously across both CNN and transformer backbones.
Comprehensive experiments, including quantitative interpretability benchmarks, user studies, and robustness tests, demonstrate that InfoDisent yields human-recognizable explanations and reasonable classification consistency.

**Strengths:**

- **S1. Scalable and sparse prototype reasoning**

The proposed method effectively addresses the scalability problem common in prototype-based XAI methods, where an excessive number of prototypes limits applicability to large datasets.
Through an orthogonal transformation that decorrelates feature channels and a deterministic max–min pooling bottleneck, it prunes redundant or non-discriminative activations, leaving only a compact set of meaningful concept channels.
This sparse and disentangled design reduces the effective number of prototypes while preserving interpretability and classification consistency.
Empirical results confirm that InfoDisent uses far fewer active channels than prior models yet maintains competitive accuracy, showing that scalability and interpretability can be achieved simultaneously.

- **S2. Comprehensive and rigorous experimental design**

The current InfoDisent framework demonstrates a clear improvement in experimental scope and rigor.
Across Sec 4 and 5 and the appendix, the authors integrate a broad range of evaluations covering classification performance, interpretability, robustness, and efficiency.
Beyond standard accuracy tests, the study includes multi-backbone benchmarking, user-study validation, and quantitative interpretability metrics using the FunnyBirds and Spatial Misalignment datasets.
Additional analyses in the appendix--such as ablation, sparsity, efficiency, and robustness tests--further confirm the model’s stability and scalability.
Together, these additions transform the evaluation into a well-rounded and reproducible experimental framework supporting the method’s reliability and generality.

**Weaknesses:**

- **W1. Moderate performance gap, backbone sensitivity, and residual background influence**

The proposed method achieves strong interpretability and sparsity, yet its classification accuracy remains slightly below that of the best-performing prototype and concept-based approaches, as reported in Tables 1 and 2.
Its performance also appears dependent on the underlying backbone: results are generally higher with ResNet, where localized convolutional priors align well with the method’s spatial disentanglement, but less competitive with transformer architectures such as ViT or Swin, which rely on global attention patterns.

Furthermore, the FunnyBirds evaluation (Fig. 7) indicates that while the model obtains the highest overall interpretability score, its Background Independence (B.I.) metric is somewhat lower than that of certain baselines. This may suggest that a portion of the learned concepts still captures contextual or texture cues inherited from the frozen backbone--such as background elements around target objects--rather than purely object-centric information.

My questions would be:
- To what extent is the observed accuracy and backbone variation driven by architectural inductive bias, with convolutional locality favoring InfoDisent’s pooling and disentanglement design?
- Could the orthogonal disentanglement mechanism interfere with globally distributed representations in ViTs?
- Does the frozen-backbone configuration limit adaptability to transformer-based features?
- Would incorporating explicit background suppression mechanisms--such as object masks, counterfactual interventions, or background-invariance regularizers--help improve B.I. while preserving sparsity and accuracy?

**Questions:**

Most of my main concerns or questions have been outlined in the Weaknesses section.

---

> ### Author Response · Authors · 2025-11-21
>
> >To what extent is the observed accuracy and backbone variation driven by architectural inductive bias, with convolutional locality favoring InfoDisent’s pooling and disentanglement design?
>
> Thank you for that question. When we observe the results across Tables 1, 2 and 3, we hypothesize that InfoDissent's performance is optimized when the backbone's bias aligns with the task's needs, demonstrating its versatility:
> Fine-Grained Tasks and CNNs: InfoDissent favors Convolutional Neural Networks (CNNs) (e.g., ResNet) because their locality bias preserves the crucial local patterns necessary for fine-grained discrimination. InfoDissent is highly effective at utilizing these local, disentangled features.
>
> ImageNet (General) Task and ViTs: InfoDissent favors Vision Transformers (ViTs) because their global aggregation bias captures the high-level, globally pooled information required for general object classification.
> In essence, InfoDissent leverages the most suitable feature representation (local for CNNs/fine-grained, global for ViTs/ImageNet) provided by the backbone to maximize performance, showing it is sensitive to the task characteristics and the inductive biases they introduce.
>
> > Could the orthogonal disentanglement mechanism interfere with globally distributed representations in ViTs?
>
> It was shown that orthogonality can lead to the improvements in ViTs representation [Huan, 2022; Tang 2022], that is why we have decided to use it. However, not any orthogonality can be successful, that is why it requires a careful crafting of this mechanism. We added a discussion in the Introduction on that.
>
> Huaibo Huang, Xiaoqiang Zhou, and Ran He. Orthogonal transformer: An efficient vision transformer backbone with token orthogonalization. In S. Koyejo, S. Mohamed, A. Agarwal, D. Belgrave, K. Cho, and A. Oh (eds.), Advances in Neural Information Processing Systems, volume 35, pp. 14596–14607, 2022.
>
> Hao Tang, Jiaxin Li, Xiaohui Li, Meng-Hui Li, Longhui Wei, Jinxing Hu, Guoliang Song, Shuo Chen, and Wu-Jun Li. O-vit: Orthogonal vision transformer. arXiv preprint arXiv:2201.12133, 2022
>
> > Does the frozen-backbone configuration limit adaptability to transformer-based features?
>
> Yes, the frozen-backbone configuration does limit adaptability to transformer-based features. This is because competing interpretable models modify and fine-tune the backbone during training, allowing them to specialize a generally-trained backbone for the fine-grained task. This adaptation leads to achieving comparable or slightly better results, especially with ViT, compared to InfoDisent, which does not modify the backbone. However, this trade-off in performance is due to the backbone non-modifying approach of InfoDisent, which greatly simplifies the training process and leads to a substantial reduction in both required computational resources and training time, as it only adjusts the last two parts of the model while leaving the original backbone unchanged.
>
> > Would incorporating explicit background suppression mechanisms--such as object masks, counterfactual interventions, or background-invariance regularizers--help improve B.I. while preserving sparsity and accuracy?
>
> Yes, we agree that incorporating explicit background suppression mechanisms is an interesting and potentially beneficial future research direction to improve Background Invariance (B.I.) for InfoDissent. The most promising mechanisms suggested are object masks or another branch to disentangle background from foreground. We believe these changes should not significantly impact sparsity or accuracy, as they only influence which parts of the image are used to derive prototypical parts. However, this would require substantial changes in the experiments and architecture, so we see it as a part of a future works.

---

### Author Response · Authors · 2025-11-28

We thank the Area Chair and Reviewers for their constructive feedback and engagement.

Below is a summary of the key points raised during the discussion period and the corresponding changes we have incorporated into the revised manuscript to address them.

Clarification of Architecture and Method

Feedback: Reviewers (FCXn, xG6k) pointed out discrepancies between the simplified architecture description (e.g., non-differentiable MaxPool) and the actual training implementation (Gumbel-Softmax).

Changes:

Gumbel-Softmax Integration: We have updated Section 3 to explicitly describe the Gumbel-Softmax estimator used during training to enable differentiability. The method description now aligns with the training procedure, clarifying that the arg max operation is approximated via Gumbel-Softmax during the learning phase.

Architecture Diagram: We have ensured the description of the architecture in Figure 2 and the text accurately reflects the sparse pooling mechanism and the information bottleneck principle.

Disentanglement and Metrics

Feedback: Concerns were raised regarding the definition of "disentanglement" and the placement of quantitative metrics (RV coefficient) in the Appendix rather than the main text (FCXn, xG6k)

Changes:

RV Coefficient in Main Text: We have moved the RV coefficient results (previously in the Appendix) to Table 4 in the main body. This table demonstrates that InfoDisent consistently achieves lower or comparable channel correlations than baselines.

Definition: We added a clearer definition of disentanglement in Section 3, emphasizing the role of orthogonal transformations and sparse pooling in enforcing independent, atomic concept channels.

Experimental Rigor and RobustnessFeedback: Reviewers requested standard deviations for performance metrics (xG6k) and validation on out-of-distribution or distinct domains like medical imaging to prove transferability (WmEX)

Changes:

Standard Deviations: We have updated Tables 1, 2, and 3 to include standard deviations for all accuracy results, ensuring statistical rigor.

Medical Dataset Experiment: We introduced a new experiment on pneumonia detection using chest X-rays in Appendix. The results in Table 17 show that InfoDisent (frozen backbone) achieves 90.55% accuracy, comparable to the 92.57% of a fully fine-tuned ResNet-18, demonstrating robustness in transfer learning.

Interpretability and Limitations

Feedback: Reviewers noted the lack of human-readable labels for prototypical parts and the trade-off between accuracy and interpretability (FCXn, 3ACv).

Changes:

Discussion on Concepts: We have expanded the Limitations section to explicitly acknowledge that while prototypical parts provide visual interpretability, they do not inherently possess human-understandable textual labels. We referenced recent work on using Vision-Language Models to bridge this gap.

Concept Bottleneck References: We added citations to relevant Concept Bottleneck literature (e.g., Koh et al., 2020; Hu et al., 2025) in Section 2 to better contextualize our work.

Ablation StudiesFeedback: Questions were asked regarding the causality of sparsity and the necessity of specific components (xG6k, WmEX).

Changes:

Sparsity Analysis: We referenced Appendix A and Table 16 which provide an ablation study on channel usage, demonstrating that InfoDisent utilizes significantly fewer channels than baseline models, confirming the efficacy of our bottleneck design.

We believe these revisions address the primary concerns regarding clarity, rigor, and scope, significantly strengthening the manuscript.

---

### Meta-Review · Area_Chair_srNe · 2026-01-08

**Summary:**

The paper has valuable points evidenced by Reviewers, but also a number of concerns; the scoring was 6, 2, 2, 4.

Raised issues include moderate performance wrt state of the art, in some cases slightly below it, and dependent from the used backbones. It is asked to clarify/discuss to what extent performance and backbone are affected by architectural inductive bias, better understanding of the disentanglement mechanism when using transformer-based models, and how Background Independence (BI) metric can be affected by masks or background suppression methods while preserving accuracy.

Other comments addressed a weak state-of-the-art analysis not fully covering the addressed topic (missing concept bottleneck models). There are also interpretability issues regarding the discovered concepts (at both local and global level), and the lack of definition (or rigorously evaluated) of information disentanglement.  No clear evidence (into the loss definition) that independent, non-overlapping atomic concepts will be extracted.

Similarly, it's not fully clear the performance-interpretability trade-off In the experimental stage, as well as some results are not all clearly commented, also missing ablations. Organization is not optimal, much important info is not in the main paper but in the appendix. Some parts of the method's description are not clear or confuse (definitions, notation, etc.), theoretical explanation of the sparsity is insufficiently reported.

Clarity and weak paper organization are comments raised by more than a Reviewer, so as the section Ablations in the Appendix, which is not presenting clear ablations, but rather how the method performs under different design options. Quality of the interpretation is debatable, i.e., similar to other prototypical part methods, and the actual advantage of the proposed method does not emerge clearly.

Similar to some of previous comments, also here it's unclear the consistency of the prototypes or concepts along classes and samples, missing loss terms forcing sparsity and independence, no cue about the transferability of the method (eg., imagenet -> medical data).

In summary many criticisms, many commonly raised by more than a reviewer.

**Reviewer Concerns:**

There are a lot of concerns, mostly addressed by authors.

It is difficult to clearly figure out if the comments about the paper organization and clarity have been fixed (even if something has been done in this sense). Interpretability of the concepts, a better definition and theoretical analysis, low-quality results do not seem to be well addressed.

In the end, a large number of comments would require a strong revision, which is difficult to assess without a further review.

**Reviewer Scores:**

Rev. FCXn engaged in a short discussion not ended, but he didn't seem so satisfied of all the authors' replies.
Also Rev. WmEX raised remarks difficult to be fixed (clarity, organization) by authors' rebuttal.

To the authors' opinion, Rev. xG6k wrote an LLM generated review, and raise a large number of of comments. I am not sure this is a generated review, but for sure comments are very punctual in some cases. However, some are similar to comments from other reviewers. He seems to accomodate authors' feedback. In the end, the paper is not bad, authors made a good job to reply to all comments, but some concerns still remain, and in any case the required revision seems too strong for an acceptance without a further revision.

---

### Decision · Program_Chairs · 2026-01-26

Reject